# The post-abscission midbody is an intracellular signaling organelle that regulates cell proliferation

Eric Peterman[1], Paulius Gibieža[2], Johnathon Schafer[1], Vytenis Arvydas Skeberdis[2], Algirdas Kaupinis[3], Mindaugas Valius [3], Xavier Heiligenstein [4], Ilse Hurbain[4,5], Graca Raposo[4,5] & Rytis Prekeris[1]

Once thought to be a remnant of cell division, the midbody (MB) has recently been shown to have roles beyond its primary function of orchestrating abscission. Despite the emerging roles of post-abscission MBs, how MBs accumulate in the cytoplasm and signal to regulate cellular functions remains unknown. Here, we show that extracellular post-abscission MBs can be internalized by interphase cells, where they reside in the cytoplasm as a membrane-bound signaling structure that we have named the MBsome. We demonstrate that MBsomes stimulate cell proliferation and that MBsome formation is a phagocytosis-like process that depends on a phosphatidylserine/integrin complex, driven by actin-rich membrane protrusions. Finally, we show that MBsomes rely on dynamic actin coats to slow lysosomal degradation and propagate their signaling function. In summary, MBsomes may sometimes serve as intracellular organelles that signal via integrin and EGFR-dependent pathways to promote cell proliferation and anchorage-independent growth and survival.

[1] Department of Cell and Developmental Biology, University of Colorado Anschutz Medical Campus, Aurora, CO 80045, USA. [2] Institute of Cardiology, Lithuanian University of Health Sciences, Kaunas 44307, Lithuania. [3] Proteomics Center, Institute of Biochemistry, Vilnius University Life Sciences Center, Vilnius University, Vilnius 10257, Lithuania. [4] Institut Curie, PSL Research University, CNRS, UMR144, Structure and Membrane Compartments, Paris 75005, France. [5] Institut Curie, PSL Research University, CNRS, UMR144, Cell and Tissue Imaging Facility (PICT-IBiSA), Paris 75005, France. Correspondence and requests for materials should be addressed to R.P. (email: rytis.prekeris@ucdenver.edu)

The final stage of cell division is abscission, the physical separation of the plasma membrane of the two daughter cells. Residing between the two daughter cells during cytokinesis is a microtubule-rich, proteinaceous structure known as the midbody (MB). The MB has been well-studied for its role in recruiting abscission-regulating proteins during cytokinesis[1–4], but its post-mitotic roles remain largely unexplored. Until recently, the MB was thought to be discarded after division by either releasing it into extracellular space or by autophagosomal degradation[5,6]. Recent studies have shown that MBs can be retained and accumulate in stem and cancer cells long after mitosis has been completed (post-abscission MBs)[7–9]. Thus, it has been proposed that post-abscission MBs function as signaling platforms that regulate cell stemness, as well as aggressiveness of cancer cells[7,8,10,11]. Additionally, it was shown that post-abscission MBs can function as a polarity cue that regulates apical lumen formation, neurite extension, and mitotic spindle orientation[12–14]. A role for post-abscission MBs has also been shown for cell fate determination in *Caenorhabditis elegans*[13,15,16]. While the functional importance of post-abscission MBs in these processes is well established, how these MBs signal remains elusive. Whether MBs actually regulate cell stemness or cancer progression also remains inconclusive, due to several conflicting studies[17–20]. Thus, the fate and function of these long-lived post-abscission MBs remains to be fully understood and is the focus of this study.

How cells retain and accumulate post-abscission MBs is poorly understood and somewhat controversial. It has been suggested that cytokinetic abscission can sometimes occur only on one side of the MB. This has led to the notion that asymmetric abscission results in one daughter cell inheriting the post-abscission MB (Fig. 1b)[2]. Alternatively, during symmetric abscission (abscission on both sides of the MB), the post-mitotic MB can be released into the extracellular space and can then be engulfed by surrounding cells (Fig. 1b)[9,17–19]. The functional outcomes of internalization versus inheritance of MBs remains unclear. Furthermore, if and how these MBs signal to regulate stemness or tumorigenicity remains to be determined.

In this study, we focus on identifying the mechanisms that regulate post-abscission MB retention, as well as functional consequences of MB accumulation. Using a transcriptome analysis approach, we demonstrate that accumulation of post-abscission MBs leads to an increase in transcription of genes that promote cell division. We show that internalization of post-abscission MBs leads to an increase in proliferation and anchorage-independent growth. Characterization of the recognition and internalization machinery revealed that MB engulfment is an active process resembling phagocytosis. The recognition of post-abscission MBs was found to be dependent a phosphatidylserine (PS)–integrin complex. We show that once internalized, MBs form membrane-bound organelles that we term MBsomes, and that these MBsomes are protected from lysosomal degradation by the formation of dynamic actin coats. Lastly, we show that MBsomes signal, at least in part, via EGF receptors (EGFRs) and αVβ3 integrins that are present in the MBsome membrane. Collectively, this study identifies a MB-dependent signaling organelle, the MBsome, and shows that MBsome signaling regulates cell proliferation and anchorage-independent growth.

## Results

**Post-abscission midbodies increase cell proliferation**. We set out to determine the function of post-abscission MBs and how/if they signal to affect cellular functions. To that end, we used a HeLa cell line stably expressing MKLP1-GFP (well-established MB marker; Supplementary Fig. 1D), allowing us to use flow cytometry to enrich for interphase cells containing MBs (+GFP-MB) and compare them to HeLa cells without post-abscission MBs (−GFP-MB) (Supplementary Fig. 1A–C). To determine whether accumulation of MBs lead to changes in overall cell fate, we compared the transcriptomes of +GFP-MB and –GFP-MB cell using mRNAseq analysis. Interestingly, the majority of up-regulated genes are known to either directly enhance cell division or regulate actin and microtubule dynamics (Fig. 1a and Supplementary Data 1). Such genes included *Ki67*, *Aurora A*, *CenpE*, *ArhGAP11b*, and *Plk1*, all of which are either known indicators of proliferation or genes that, when overexpressed, promote cell proliferation (for qPCR validation see Supplementary Fig. 1E).

While our transcriptome analysis suggested that post-abscission MB accumulation increases expression of genes involved in proliferation, it did not rule out the possibility that we were arbitrarily selecting for populations that were stochastically dividing faster, thus accumulating more MBs through rapid divisions. To address this, we devised a protocol to purify post-abscission, extracellular MBs that are released via symmetric abscission from HeLa cells expressing MKLP1-GFP (Fig. 1b, Supplementary Fig. 2A, B). Since it has previously been shown that one of the daughter cells can internalize the MB immediately after cell division[9], we tested whether purified MBs in the media can be taken up by other interphase cells. We added purified MKLP1-GFP containing MBs (GFP-MB) to HeLa cells and incubated for 3 h, followed by washing with media to remove any unbound GFP-MBs. HeLa cells readily internalized purified GFP-MBs (Fig. 1c, Supplementary Fig. 2D). Furthermore, cells were able to internalize multiple post-abscission MBs. From all GFP-MB containing cells, 55.8% had 1 MB, 22.1% had 2 MBs, 12.6% had 3 MBs, and 9.6% had 4 MBs ($n = 95$). Other cell types, such as MDCK cells and MDA-MB-231 cells, were also capable of internalizing GFP-MBs purified from HeLa cells (Supplementary Fig. 2E and F).

The ability of cells to internalize large extracellular objects is usually driven by phagocytosis-like mechanisms and it is now well-established that these phagosome-like organelles usually fuse with lysosomes to rapidly degrade cargo[21,22]. To test whether GFP-MBs taken up by interphase cells are rapidly degraded, we incubated 'fed' cells for varying amount of times and stained them with antibodies against the lysosomal/late endosomal marker CD63. As expected, some GFP-MBs were surrounded by CD63-containing membrane (Fig. 1c), but even after 24–48 h post 'feeding' we could still observe MBs that were not encompassed by CD63+ organelles, indicating that they have not been degraded up until that point (Fig. 1c, d). To further determine whether internalized MBs can be retained by interphase cells, we incubated HeLa cells for 3 h with purified GFP-MBs and then followed the fate of these MBs for 16 h using time-lapse microscopy. While some internalized MBs were degraded after a few hours post-internalization (Supplementary Fig. 2G, see MB#2), 57% of internalized GFP-MBs ($n = 14$) could still be observed after 16 h post-incubation (Supplementary Fig. 2G, see MB#1).

The ability to 'feed' interphase cells with MBs allowed us to directly test whether post-abscission MBs can regulate cell proliferation as suggested by our transcriptome analysis. We 'fed' HeLa cells with GFP-MBs and then analyzed their proliferative capacity. First, we incubated 'fed' cells for 24 h and then stained for Ki67, a classical marker for proliferation. In comparison to −GFP-MB cells (cells on the same coverslip that did not internalize MBs), cells that contained two or more MBs had significantly higher Ki67 levels (Fig. 2a). Next, we stained with anti-acetylated tubulin antibodies to assess how many cells within each population are undergoing mitosis. We found that a higher fraction of GFP-MB-containing cells were in mitosis as

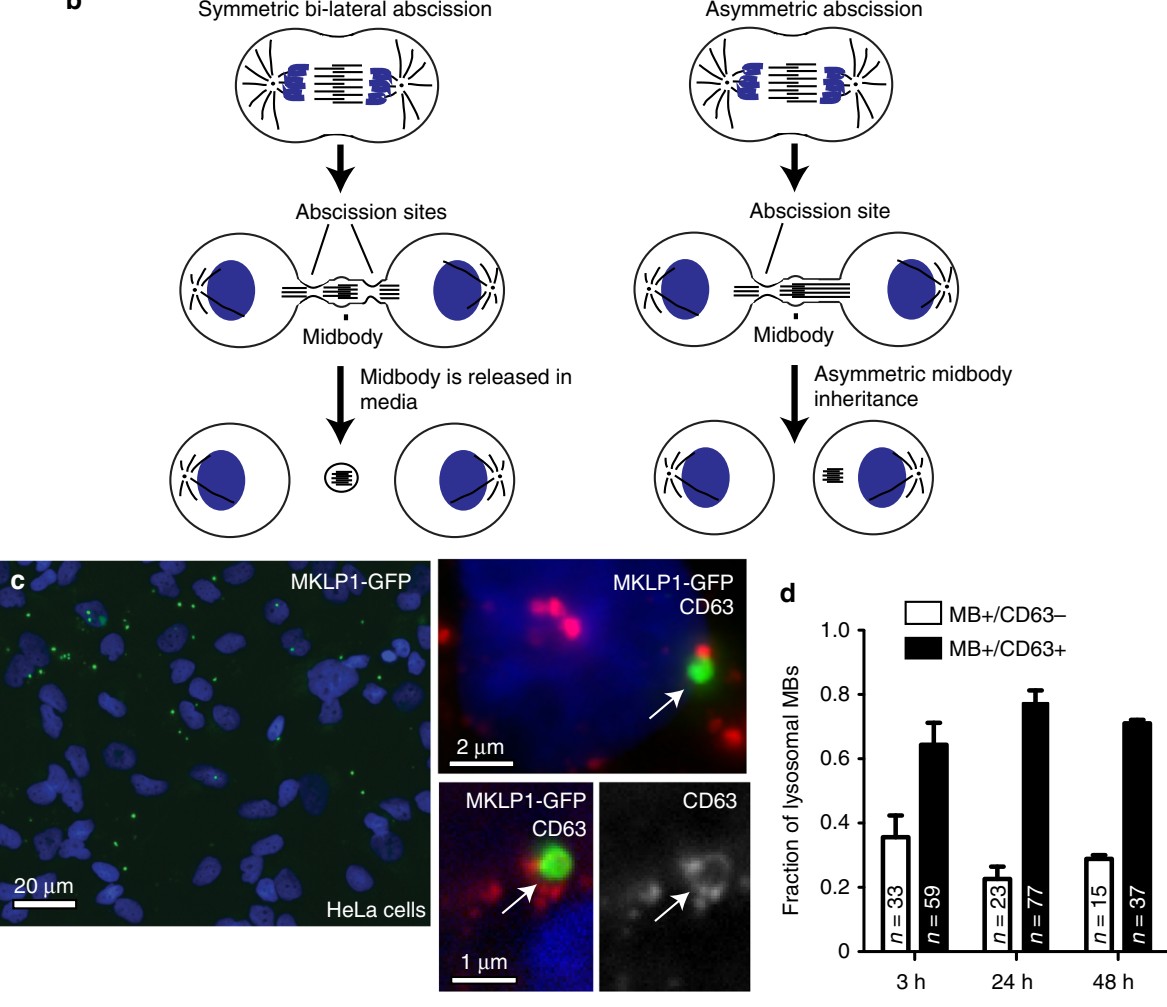

| Annotation cluster 1 | Enrichment score: 7.487 | Count (Total) | % Total | *P* value |
|---|---|---|---|---|
| | Term: Mitosis | 16(61) | 26.23 | 9.86E−17 |
| | Term: Mitotic nuclear division | 14(61) | 22.95 | 1.42E−14 |
| | Term: Cell division | 16(61) | 26.23 | 3.36E−14 |
| | Term: Cell cycle | 18(61) | 29.51 | 2.82E−13 |
| Annotation cluster 2 | Enrichment score: 6.198 | | | |
| | Term: Centromere | 9(61) | 14.75 | 2.16E−09 |
| | Term: Kinetochore | 8(61) | 13.11 | 6.18E−09 |
| Annotation cluster 3 | Enrichment score: 4.105 | | | |
| | Term: Mircotubule | 9(61) | 14.75 | 5.54E−07 |
| | Term: Cytoskeleton | 15(61) | 24.59 | 7.28E−07 |

**Fig. 1** Post-abscission MBs are internalized and accumulate in interphase cells. **a** DAVID functional cluster analysis of mRNAs up-regulated in GFP-MB+ cells. *P*-value is represented by modified Fisher exact *P*-value. **b** Schematic representation of asymmetric and symmetric cell abscission. **c, d** HeLa cells were incubated with purified GFP-MBs for 3 h. Cells were then washed and incubated for additional 3 h before fixation and staining with anti-CD63 antibodies. In **c** arrow in top inset shows MB that does not colocalize with CD63, while arrow in bottom insets shows MB that is surrounded by CD63-positive membrane. Panel **d** shows quantification of the colocalization between GFP-MBs and CD63. *n* is the number of internalized MBs counted for each condition. Three biological replicates were used to obtain data

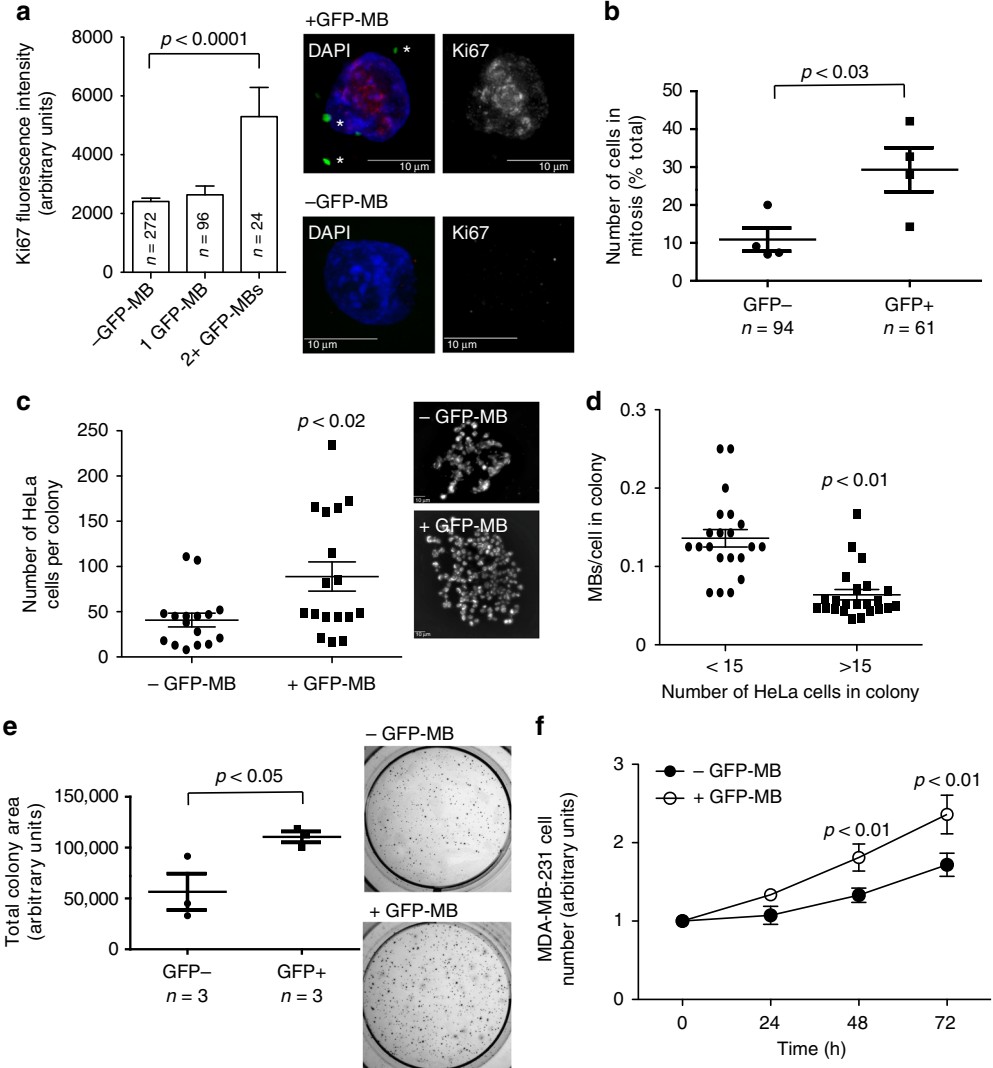

**Fig. 2** Post-abscission MB internalization stimulates cell proliferation. **a, b** HeLa cells were incubated with purified GFP-MBs for 3 h, followed by wash and incubation for another 24 h. Cells were then fixed and stained with anti-Ki67 (**a**) or anti-acetylated tubulin (**b**). Panel **a** shows quantification of Ki67 levels in each cell, while panel **b** shows quantification of the percentage of cells in mitosis (as determined by the presence of mitotic spindle). Data shown are the means and standard deviations derived from 200 randomly chosen cells from three independent experiments for Ki67 staining and four independent experiments for mitotic index quantification (Student's unpaired, two-tailed t-test). Asterisks in images (**a**) mark post-abscission MBs. n shows the number of cells analyzed for each condition. **c** Hela cells stably expressing mCherry-CAAX were fed MBs and GFP-MB+ cells were identified by fluorescence microscopy. Unfed cells were used as a control. Cells were tracked using glass bottom dishes and were then tested for their proliferative capacity by imaging the same cell 7 days post feeding. Data shown are the means and standard deviations derived from four independent experiments (Student's unpaired, two-tailed t-test). **d** HeLa cells were plated at low density and grown for 72 h to form colonies. Cells were then incubated for 3 h with purified GFP-MBs. The ability to internalize MBs by fast dividing cells (formed large colonies) was then compared to slow dividing cells (formed small colonies). Data shown are the means and standard deviations derived from three independent experiments (Student's unpaired, two-tailed t-test). **e** Hela cells with internalized GFP-MBs were isolated by flow sorting. Cells were then tested for their ability to grow in soft agar. Data shown are the means and standard deviations derived from three independent experiments (Student's unpaired, two-tailed t-test). **f** MDA-MB-231 cells with internalized GFP-MBs were isolated by flow sorting. Cells were then tested for their proliferative capacity. Data shown are the means and standard deviations derived from three independent experiments (Student's unpaired, two-tailed t-test)

compared to control cells that were present on the same coverslip but did not internalize the GFP-MBs (Fig. 2b). Finally, we did flow-cytometry-based cell cycle analysis showing that cells with GFP-MBs had a higher G2/M to G1 ratio (Supplementary Fig. 3G). These data indicate that GFP-MB internalization stimulates proliferation of HeLa cells.

We next wanted to determine whether accumulation of post-abscission MBs can also lead to an increase in proliferation. HeLa cells expressing mCherry-CAAX were seeded at a low density on glass-bottom dishes and fed with GFP-MBs. Individual cells with

or without GFP-MBs were identified and colony size was assessed after 7 days. Consistent with the involvement of MBs in regulating proliferation, colony size that originated from GFP-MB-containing cells was greater than the controls (cells that did not contain GFP-MB; Fig. 2c). Internalization of GFP-MBs also increased proliferation of MDA-MB-231 cells, ensuring these effects are not limited to HeLa cells (Fig. 2f). To test for tumorigenicity, GFP-MB fed cells were flow sorted into populations with GFP-MBs (+GFP-MBs) or without (−GFP-MBs), and then plated into soft agar to test for anchorage

independent growth. We found that 14 days post-embedding +GFP-MBs had larger and more numerous colonies than −GFP-MBs (Fig. 2e, Supplementary Fig. 5).

Our data suggest that internalization of MBs leads to an increase in proliferative capacity. However, an alternative possibility is that faster dividing cells may be internalizing MBs more readily. To test for this, we plated cells at a very low density and allowed them to form colonies. In this assay, larger colonies represent a faster dividing sub-population of HeLa cells, while small colonies represent slower dividing HeLa cells. These colonies were then fed GFP-MBs, and MB internalization rates were assessed. As shown in Fig. 2d, cells in the large colonies (fast dividers) internalized MBs at a lower efficiency. We next tested whether internalization of any large extracellular objects (rather than specifically MBs) may lead to increased proliferation. We incubated HeLa cells with fluorescent *Escherichia coli* BioParticles. As shown in Supplementary Fig. 2C, BioParticle internalization did not recapitulate the MB-induced increase in proliferation. Finally, we tested whether +GFP-MB cells retain higher proliferation rates after MBs are degraded. To determine this, we incubated HeLa cells with purified GFP-MBs, followed by flow-sorting cells into +GFP-MB and −GFP-MB populations. Cells were cultured for 7 days to ensure degradation of MBs, followed by measurement of proliferation. As shown in Supplementary Fig. 3B there were no differences in proliferation rates suggesting that cells revert to the original proliferation state after internalized MBs are degraded. Furthermore, there were also no differences in the internalization of purified GFP-MBs applied to both populations of cells after 7-day incubation (Supplementary Fig. 3A).

To further confirm that the internalization of post-abscission MBs lead to increase in mRNAs necessary for proliferation, we next incubated HeLa cells with purified GFP-MBs followed by flow sorting 24 h later. Cells with or without internalized GFP-MBs were then analyzed by mRNAseq. Consistent with our hypothesis that internalization of post-abscission MBs leads to stimulation of proliferation, a large subset of genes known to be required for proliferation were upregulated (Supplementary Data 5A). Importantly, genes identified and validated in our first RNAseq analysis were also increased in cells containing internalized GFP-MBs (Supplementary Data 5B). Collectively, these data show that internalized post-abscission MBs may be signaling organelles that can regulate cell proliferation and tumorigenic potential. In the remainder of the study, we will be referring to these signaling organelles as MB-containing signaling endosomes or MBsomes.

**Actin-rich protrusions mediate MB internalization**. Midbodies released into extracellular space during symmetric abscission contain the membrane envelope that originated from the plasma membrane of the mother cell (Fig. 1b)[9,16,23]. Consequently, cells could internalize extracellular MBs by fusing plasma membrane with the MB membrane envelope or by engulfing MBs via a regulated, phagocytosis-like process, thereby creating double membrane-bound MBsomes. To test both possibilities, we analyzed GFP-MB internalization using HeLa cells stably expressing mCherry-CAAX. As shown in Fig. 3, GFP-MBs are internalized via the formation of plasma membrane-derived protrusions (see arrows in Fig. 3a). To better visualize these membrane protrusions, we incubated purified GFP-MBs with HeLa cells and then analyzed plasma membrane bound MBs by high-resolution 3D tomography. As shown in Fig. 3b, c, post-abscission MBs are bound to the plasma membrane by attaching to multiple membrane protrusions.

While our data demonstrates that post-abscission MBs are internalized by a phagocytosis-like mechanism rather than directly fusing with plasma membrane, it is still possible that

the membranous MB envelope can later fuse with endocytic membranes, thus releasing MBs into cytoplasm. To test for this, we incubated mCherry-CAAX-expressing cells with GFP-MBs, followed by wash and an additional 24 h incubation. As shown in Supplementary Fig. 3C, even 24 h post-internalization we could still observe mCherry-CAAX containing membrane surrounding the MB (Supplementary Fig. 3E). Some of these membrane-bound MBs are targeted for degradation, since the surrounding mCherry-CAAX membrane also contained the lysosomal marker CD63 (Fig. 1c, d). However, consistent with our previous data, even 48 h post-feeding there is a fraction of MBs that is surrounded by mCherry-CAAX membranes but do not stain for CD63 (Fig. 1c, d and Supplementary Fig. 3C).

Since GFP fluorescence can be rapidly quenched in acidic late endosomes/lysosomes, it is possible that we underestimate the number of MBs that are being degraded. To that end, we fed HeLa cells with GFP-MBs and stained cells with anti-CD63 and anti-GFP antibodies 24 h later. Surprisingly, we could detect GFP fluorescence in all internalized MBs, even if they were located within CD63-positive late endosomes/lysosomes (Supplementary Fig. 4A). This data again suggests that MBs retain membranous envelope that is left over from dividing cell for at least 24 h and likely protects MKLP1-GFP from acidification.

To further confirm that post-mitotic extracellular MBs were encased in membrane remaining from division, MBs were isolated from mCherry-CAAX-expressing cells and stained with anti-acetylated tubulin antibodies. We were able to show that mCherry-CAAX clearly encompasses purified tubulin-rich MBs (Supplementary Fig.3D). Lastly, we created a HeLa cell line that expressed both mCherry-CAAX and MKLP1-GFP. MBs were isolated from these cells and 'fed' onto normal HeLa cells; we observed that the MKLP1-GFP signal was enveloped by mCherry-CAAX (Supplementary Fig. 3E).

To better localize internalized MBs, we next used correlative light and electron microscopy (CLEM). To that end, we 'fed' HeLa cells with purified GFP-MBs and fixed cells by high-pressure freezing 24 h later. The samples were prepared to preserve fluorescence after resin embedding for targeted thin section electron microscopy. Fluorescence was used to identify cells that have internalized MBs. We could observe internalized MBs that appear to be surrounded by membranes of the engulfing cell, as well as the membrane from the daughter cell division (Fig. 3d–g). At some parts of these MBs, membranes were separated far enough to see that the membranes surrounding internalized MBs appear to consist from the outside membrane (likely derived from internalizing cell) and internal membrane (likely the MB-associated membrane) (Fig. 3e–g; see black and white arrows). Taken together, these experiments suggest that extracellular post-abscission MBs are enveloped by membrane originating from the dividing cell and, once internalized, becomes further enveloped by second membrane originating from the cell that has engulfed it.

Formation of phagosome-like plasma membrane protrusions is known to be driven by Rac1-dependent localized actin polymerization[22,24–26]. Previous studies have suggested that MB internalization by one of the daughter cells involves actin polymerization[9]. Additionally, recent work in *C. elegans* embryos suggest that the Rac1 is required for MB internalization and degradation[16]. Thus, we next tested whether actin mediates GFP-MB engulfment and MBsome formation by using HeLa cells expressing LifeAct-mCherry. As shown in Fig. 4a, b, actin protrusions and rosettes can be readily observed at the sites of GFP-MB internalization. Similar dynamic actin rosettes could also be observed at the site of GFP-MB internalization by time-lapse microscopy (Fig. 4c). Importantly, pre-incubation of cells with Rac1 inhibitor or cytochalasin D decreased MB

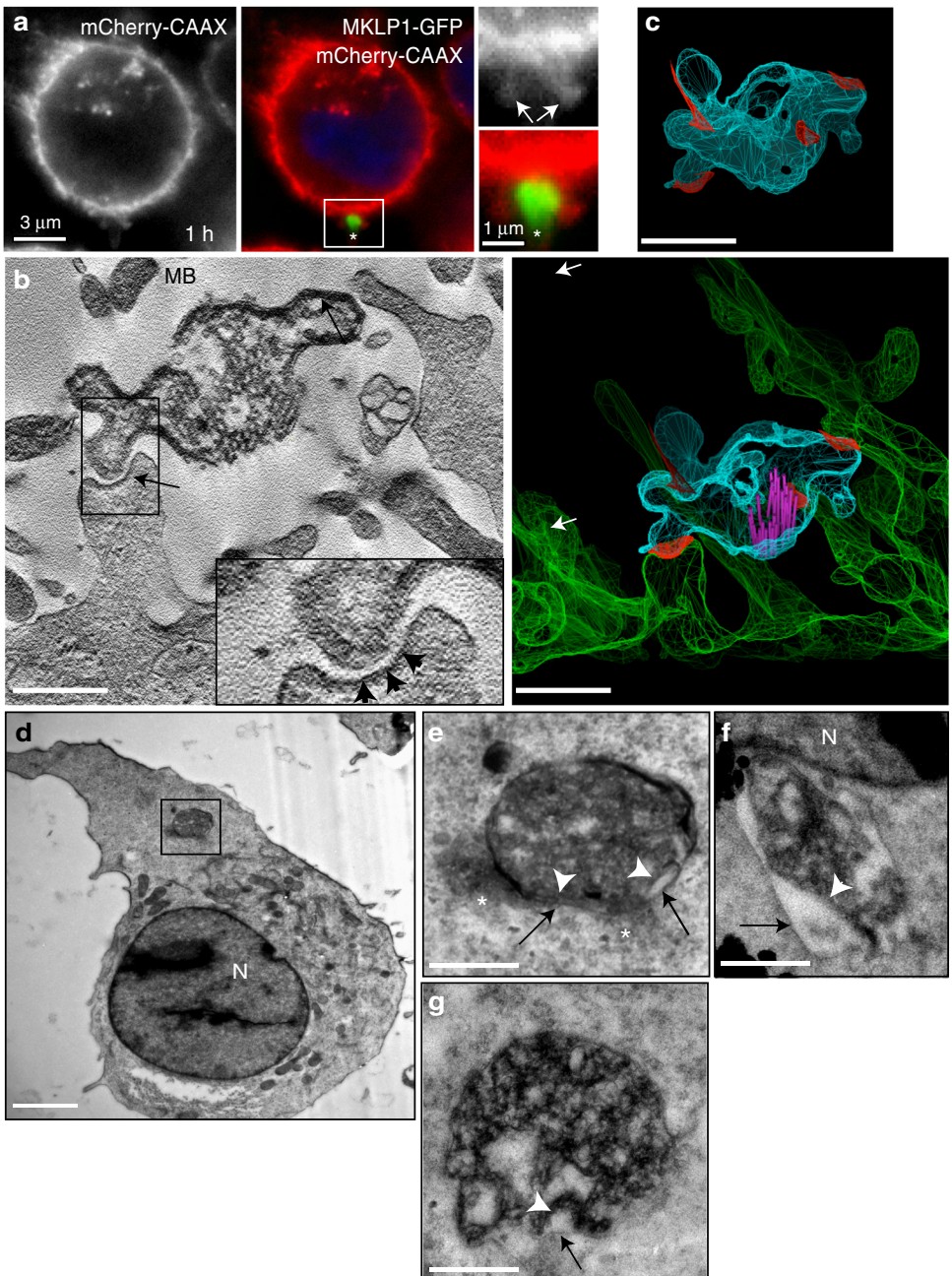

**Fig. 3** Extracellular MBs are internalized by phagocytosis-like mechanism. **a** HeLa cells stably expressing mCherry-CAAX were incubated with GFP-MBs for 3 h followed by washing and incubation with media at 37 °C for another hour. Cells were then fixed and analyzed by microscopy. Arrows point to membranous protrusions surrounding MB during internalization. **b, c** HeLa cells were incubated with GFP-MBs for 3 h. Cells were then washed and incubated for another 3 h, followed by tomography analysis. Arrow in **b** (left image) point to the MB. Arrowheads in **b** point to electron dense coat associated with sites where MB binds to internalizing cell plasma membrane. Right panel in **b** shows plasma membrane and MB surface model rendered based on tomography. Panel in **c** shows MB-only surface model. Red color marks MB and plasma membrane interaction sites. Scale bars are equivalent to 500 nm. **d–g** HeLa cells were incubated with GFP-MBs for 3 h. Cells were then washed and incubated for another 24 h, followed by CLEM/thin section EM analysis. Panel **e** shows higher magnification image of MB boxed in image **d**. Panels **f** and **g** shows MBs from other cells. N-marks nucleus. Black arrow points to outside membrane. White arrow points to internal membrane. Asterisks mark actin-like electron-dense patches associated with internalized MBs. Scale bar is equivalent to 1 μm. Scale bars in inset and magnified images is equivalent to 250 nm

internalization (Fig. 4d), further confirming that Rac1-induced actin polymerization is needed for post-abscission MB internalization by HeLa cells (Fig. 4e).

**PS and integrins mediate post-abscission MB engulfment.** We next interrogated the machinery necessary for MB recognition

and internalization. To identify the MB proteins that may mediate this process, we performed proteomic analysis of purified post-abscission MBs (Supplementary Data 2). A previously published MB proteome used Triton-X 100 to extract MBs from dividing cells, thereby losing the membranous envelope in post-abscission MBs[27]. We analyzed the proteome of post-abscission MBs purified from media in the absence of

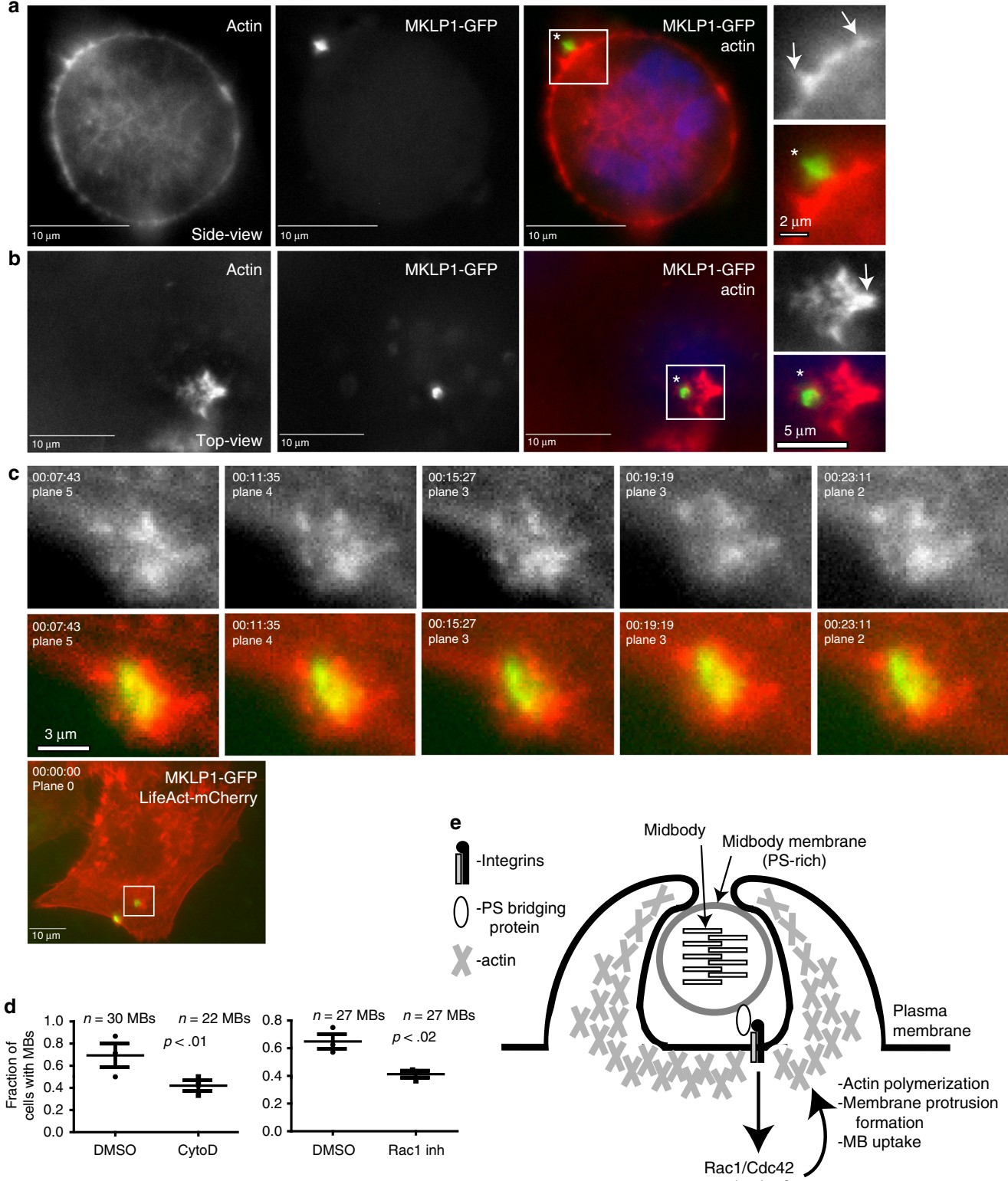

**Fig. 4** Actin polymerization mediates phagocytic MB internalization. **a–c** HeLa cells stably expressing LifeAct-mCherry were incubated with GFP-MBs for 3 h. Cells were then washed and analyzed by microscopy. Arrows in **a** and **b** point to actin-rich protrusions surrounding MBs during internalization. **c** Shows time-lapse series of actin rosette dynamics during MB internalization. **d** HeLa cells stably expressing LifeAct-mCherry were incubated with GFP-MBs for 3 h in the presence or absence of 10 μm of cytochalasin D or 100 μm of Rac1 inhibitor. Cells were then washed and incubated for another 24 h (without cytochalasin D or Rac1 inhibitor). The number of cells with internalized post-mitotic MBs were determined by fluorescence microscopy. The data shown are means and standard deviations derived from three independent experiments. $n$ shows the number of cells analyzed for each experimental condition (Student's unpaired, two-tailed $t$-test). **e** Schematic representation of proposed mechanism for MB internalization

detergents (Supplementary Fig. 2A). As expected, proteomic analysis identified many proteins known to be present at the MBs, such as tubulin, PLK1, MKLP1, Rab11, Rab35, and ESCRT-III (Supplementary Data 2). Importantly, our MB proteomic analysis also identified three proteins that are known to mediate phagocytosis, namely CCN1, EDIL3, and MFG-E8[28–30]. The commonality between these three proteins is that they all can act as molecular bridges by binding to PS and integrins and are sometimes referred to as matricellular receptors. PS, when expressed on the outer leaflet of apoptotic cells, can act as an 'eat-me' signal for surrounding phago-cytes[31,32]. Simultaneous binding of matricellular receptors to PS (on apoptotic bodies) and integrins (on phagocytes) allows recognition of apoptotic bodies, as well as activation of actin polymerizing machinery that results in engulfment of the apoptotic body[28,30,33]. Importantly, PS was found to be enriched at the MB during cytokinesis although it is unclear whether it was on the inner or outer plasma membrane leaflet[23]. We used a purified recombinant GST-tagged PS-binding (PS-BD) fragment from MFG-E8 (PS-BD; Fig. 5a) to examine if PS is found on the outer leaflet of extracellular MBs. As shown in Fig. 5f, PS is present in the outer leaflet of membranous envelope in extracellular purified GFP-MBs. We next tested whether PS is enriched in the outer leaflet of MB membrane in cells. Cells were incubated with GST-PS-BD, washed, fixed, and stained with anti-GST and anti-acetylated tubulin antibodies (to visualize central spindle and MBs) (Fig. 5g). Remarkably, we found that post-abscission extracellular MBs have high levels of PS in their outer leaflet (Fig. 5g; asterisk) as compared to the MBs during telophase MBs (Fig. 5g; arrow). While the mechanisms that mediate the increase in PS in the outer leaflet of post-abscission MBs remains unclear, our data suggests that post-abscission MBs probably need to 'mature' to allow flipping of PS to the outer leaflet in order to become internalization-competent. Next, we tested whether MFG-E8 is present in post-abscission MBs. Consistent with our proteomics data, MFG-E8 is enriched in purified post-abscission MBs as compared to cell lysate (Supplementary Fig. 2B). As shown in Supplementary Fig. 3F, we also found MFG-E8 to be enriched in MB and plasma membrane contact sites, suggesting it may act as a matricellular receptor for MB recognition.

To directly test whether MFG-E8, PS, and integrins play roles in MB recognition and engulfment, we took two approaches. First, we tested the necessity of PS for MB recognition by using recombinant GST-PS-BD derived from MFG-E8 (Fig. 5a). Since this MFG-E8 peptide is capable of binding to PS (on the MB envelope) but not integrins (on HeLa plasma membrane) it should inhibit MB binding and internalization. To test that, purified GFP-MBs were pre-incubated with recombinant GST-PS-BD and then 'fed' to HeLa cells. To measure the binding of GFP-MBs to plasma membrane, cells were analyzed at 3 h post feeding. To quantify the number of internalized MBsomes, cells were analyzed at 24 h post feeding. As shown in Fig. 5b, e, GST-PS-BD inhibited both GFP-MB binding to HeLa plasma membrane, as well as internalization and formation of MBsomes, suggesting that MB-associated PS plays a role in extracellular MB recognition and engulfment.

Our second approach was to administer cilengitide (a cyclic RGD peptide) to cells before GFP-MB feeding. Previously published work has shown that cilengitide preferentially binds to αvβ3 and αvβ5 integrin pairs[34,35]. These integrin pairs are also known to bind CCN1, EDIL3, and MFG-E8, the matricellular receptors identified in MB proteome. To first confirm the effectiveness of cilengitide in blocking αvβ3 and αvβ5 integrins, we pre-treated HeLa cells with either cilengitide (referred to as RGD) or an inactive control RGE peptide. Cells were then plated

onto fibronectin, which relies on the integrin pairs αvβ3 and αvβ5, among others, to facilitate cell adhesion[35]. Cells treated with RGD were less efficient at binding to fibronectin in comparison to RGE controls (Fig. 5c). To assess the role of αvβ3 and αvβ5 integrins on GFP-MB recognition and engulf-ment, cells grown on collagen-coated cover slips were pre-treated with RGD or RGE peptide control and incubated with purified GFP-MBs. Once again cells were analyzed after 3 or 24 h post feeding to test GFP-MB binding to plasma membrane and internalization, respectively. As with GST-PS-BD, incubation with RGD peptide inhibited GFP-MB binding to plasma membrane (Fig. 5d), as well as internalization and formation of MBsomes (Fig. 5e).

**Actin regulates MBsome degradation and signaling**. It remains unclear how some MBsomes evade degradation after inter-nalization. It is now well established that phagosomes usually fuse with lysosomes to degrade their contents. While the molecular machinery delaying the degradation of the MBsomes remains unknown, we observed that MBsomes are often (48% of inter-nalized MBs) associated with actin patches (Fig. 6a).These actin patches are not just a property of HeLa cells, since similar patches could be observed in MDA-MB-231 and 293T cells (Sup-plementary Figs. 2E and 4B). These actin patches are often asymmetric and sometimes have long actin tails protruding off the side of the MBsome (Fig. 6b, Supplementary Fig. 4B). Fur-thermore, time-lapse imaging of HeLa cells expressing LifeAct-mCherry that were fed with GFP-MBs shows that these actin patches and/or tails are very dynamic and constantly polymerize/depolymerize at the surface of the MBsome (Fig. 6c, arrows). Dynamic formation of these asymmetric actin patches correlates with MBsome movement within the cytoplasm of the cell (Fig. 6d and Supplementary Movie 3). Interestingly, electron-dense actin-like patches can also be observed near MBsomes identified by CLEM (Fig. 3e, asterisks).

It has been shown that some species of bacteria are able to hijack host cell machinery (such as Rac1 or Cdc42) and create protective actin coats around the phagosome to prevent lysosomal degradation[25]. In some instances, these bacteria are able to move throughout the cell using actin-dependent machinery[36]. Additionally, it was recently shown that similar highly dynamic actin coats can also form around beads phagocytosed by macrophages and that the formation of these actin coats inhibits phagosome and lysosome fusion[37]. Thus, we interrogated whether the loss of these actin patches might result in increased MBsome degradation. To that end, we 'fed' HeLa cells with GFP-MBs, followed by wash and incubation for 20 h to allow GFP-MB internalization and MBsome formation. Cells were then treated for 30 min with either DMSO or Latrunculin A, an actin depolymerizing agent (LatA). (Fig. 6e). Cells were then washed and incubated for another 8 h, followed by fixation. Consistent with a role for actin in protecting MBsomes from degradation, LatA treatment decreased the number of MBsomes (Fig. 6e) and increased the number of MBs associated with CD63-positive late endosomes/lysosomes (Fig. 6e). We found that LatA treatment could abrogate the effects of MBsome-induced proliferation, due to the increased degradation of MBsomes (Fig. 6f).

**EGFR and αVβ3 integrins mediate MBsome signaling**. Our remaining questions pertain to how these MBsomes signal to regulate cell proliferation. Since internalized post-abscission MBs are surrounded by two membranes, the signaling could involve some sort of transmembrane protein. We have already deter-mined that αvβ3 and/or αvβ5 integrins are important for

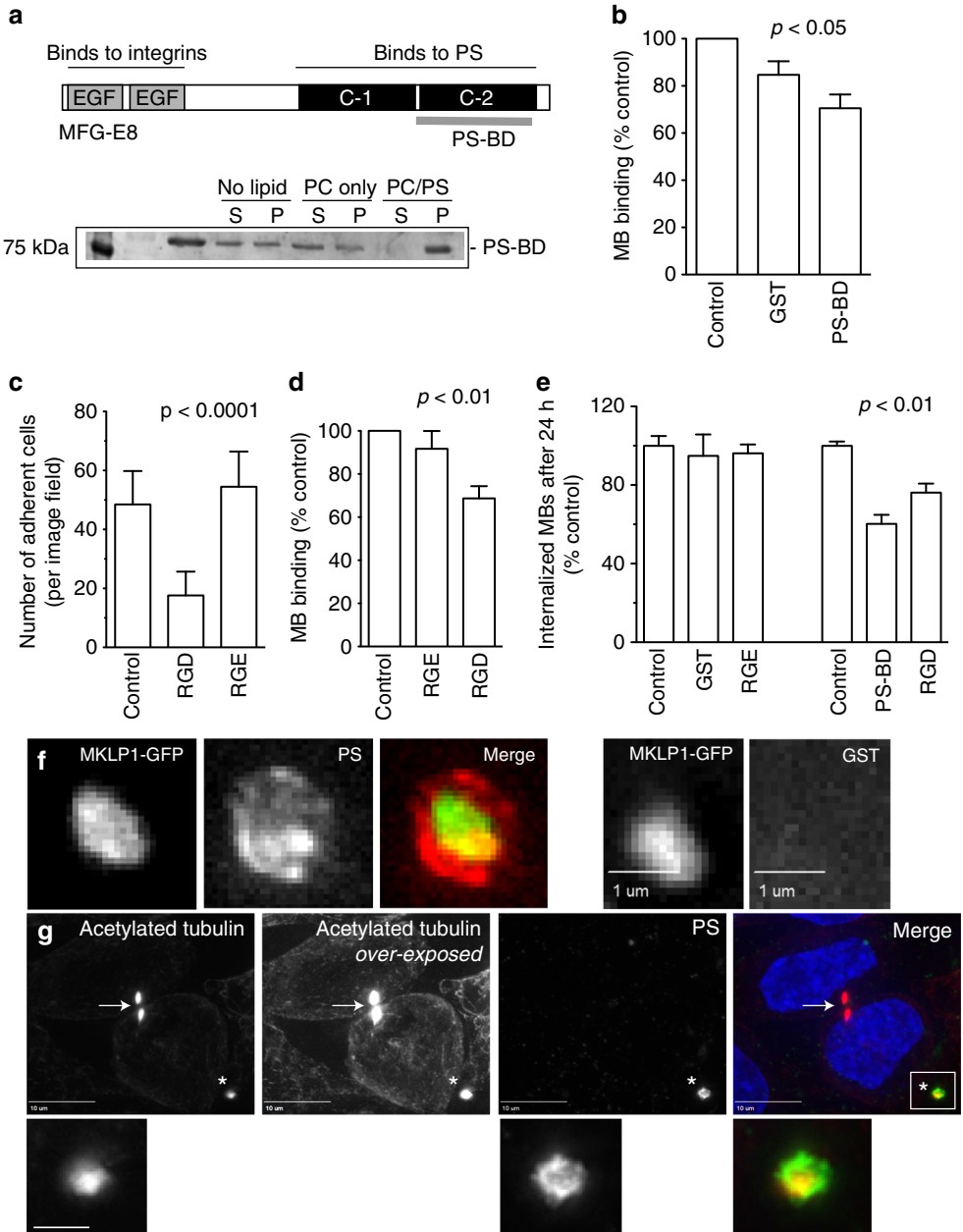

**Fig. 5** PS, matricellular receptors, and integrins mediate MB binding and internalization. **a** Recombinant PS-binding domain from MFG-E8 (PS-BD) was purified and incubated with liposomes with or without PS. Liposomes were then sedimented by centrifugation and levels of bound PS-BD determined by Coomassie staining. **b**, **d**, **e** HeLa cells or purified MBs were pre-incubated with either recombinant GST, PS-BD, or RGD cyclic peptide and its associated negative control RGE. Cells were then incubated with MBs for 3 h, followed by wash and incubation for either 3 h (**b** and **d**; to measure MB binding) or 24 h (**e**, to measure MB internalization). Cells were then fixed and number of MBs counted. Data shown are the means and standard deviations derived from three independent experiments (one-way ANOVA). **c** HeLa cells were incubated in suspension with RGD or its inactive control RGE. Cells were then plated on fibronectin-coated coverslips. The number of adhered cells were then counted. Data shown are the means and standard deviations derived from three independent experiments (one-way ANOVA). **f** Purified GFP-MBs were incubated with recombinant purified either GST-PS-BD (left panels) or GST alone (right panels). MBs were then sedimented by centrifugation, washed, fixed, and stained with anti-GST antibodies (one-way ANOVA). **g** HeLa cells were incubated with recombinant purified GST-PS-BD resuspended in serum-supplemented media. Cells were then washed, fixed and stained with anti-acetylated tubulin antibodies. Arrow in images point to mitotic MB. Asterisk marks extracellular, post-mitotic MB. Boxed region marks the part of the image shown in higher magnification insets below. Scale bar in inset is equivalent to 500 nm

internalization of post-abscission MBs, thus, we next questioned whether these integrins are involved in MBsome signaling. First, we tested whether integrins are present in MBsomes using anti-αvβ3 or anti-αvβ5 integrin antibodies. Even 24 h after MB internalization, the MBsomes still contained αvβ3 but not αvβ5 integrins (Fig. 7a, b). To determine whether internalized αvβ3

integrins are still activated, we stained cells with antibodies against phospho-FAK, a known downstream effector of the integrin-signaling pathway. As shown in Fig. 7c, d, MBsomes co-localized with phospho-FAK, indicating that even 24 h after internalization αvβ3 integrins could still be signaling from MBsomes.

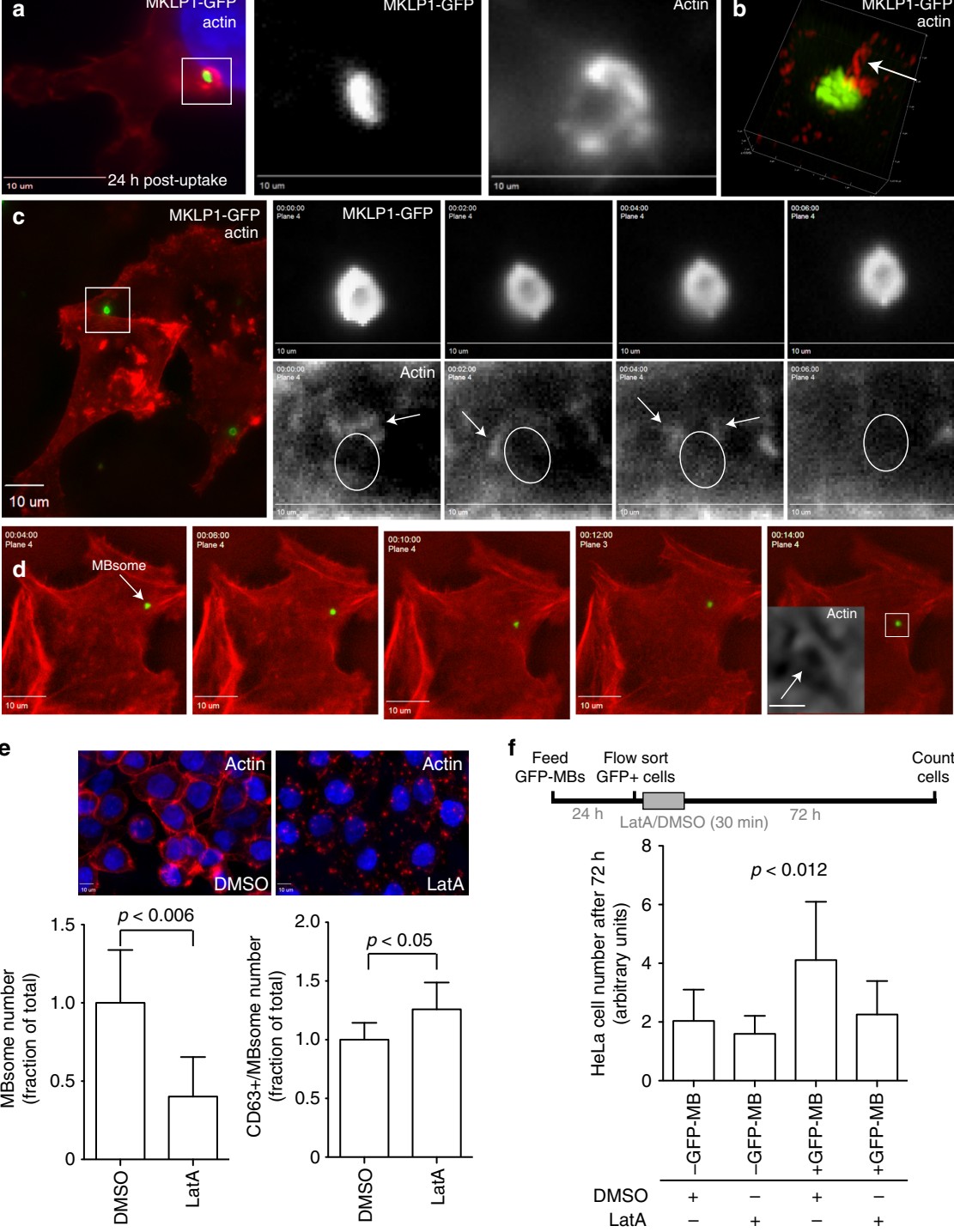

**Fig. 6** MBsome-associated actin coats inhibit MBsome degradation. **a**, **b** HeLa cells were incubated with purified GFP-MBs for 3 h. Cells were then washed and incubated for another 24 h, followed by fixation and staining with Alexa568-phalloidin. Cells were then imaged using either a regular epifluorescence microscope (**a**) or a SIM super-resolution microscope (**b**). Arrow in **b** points to MBsome associated actin patch. **c**, **d** Cells expressing mCherry-LifeAct were incubated with purified GFP-MBs. Cells were then washed and incubated for another 24 h. The dynamics of MBsome associated actin (**c**) or intracellular mobility of the MBsome (**d**) was analyzed by time-lapse microscopy. Arrows in **c** points to MBsome associated actin. Arrows in **d** point to MBsomes. Box in **d** marks the part of the image that was used for the inset at the last time-point of time-lapse series. Scale bar in box **d** is equivalent to 1 μm. **e** HeLa cells were flow sorted for the presence of internalized GFP-MBs 24 h post feeding. Cells were then treated with either DMSO or Latrunculin A for 30 min, followed by washing and incubation for another 8 h. Cells were then fixed and analyzed for the presence of MBsomes (left bar graph) and for CD63+ MBsomes (right graph). Data shown are the means and standard deviations derived from five independent experiments (Student's unpaired, two-tailed *t*-test). **f** HeLa cells were flow sorted for the presence of internalized GFP-MBs 24 h post feeding. Cells were then treated with either DMSO or Latrunculin A for 30 min, followed by washing and incubation for another 72 h to test for cell proliferation. Data shown are the means and standard deviations derived from six independent experiments (one-way ANOVA)

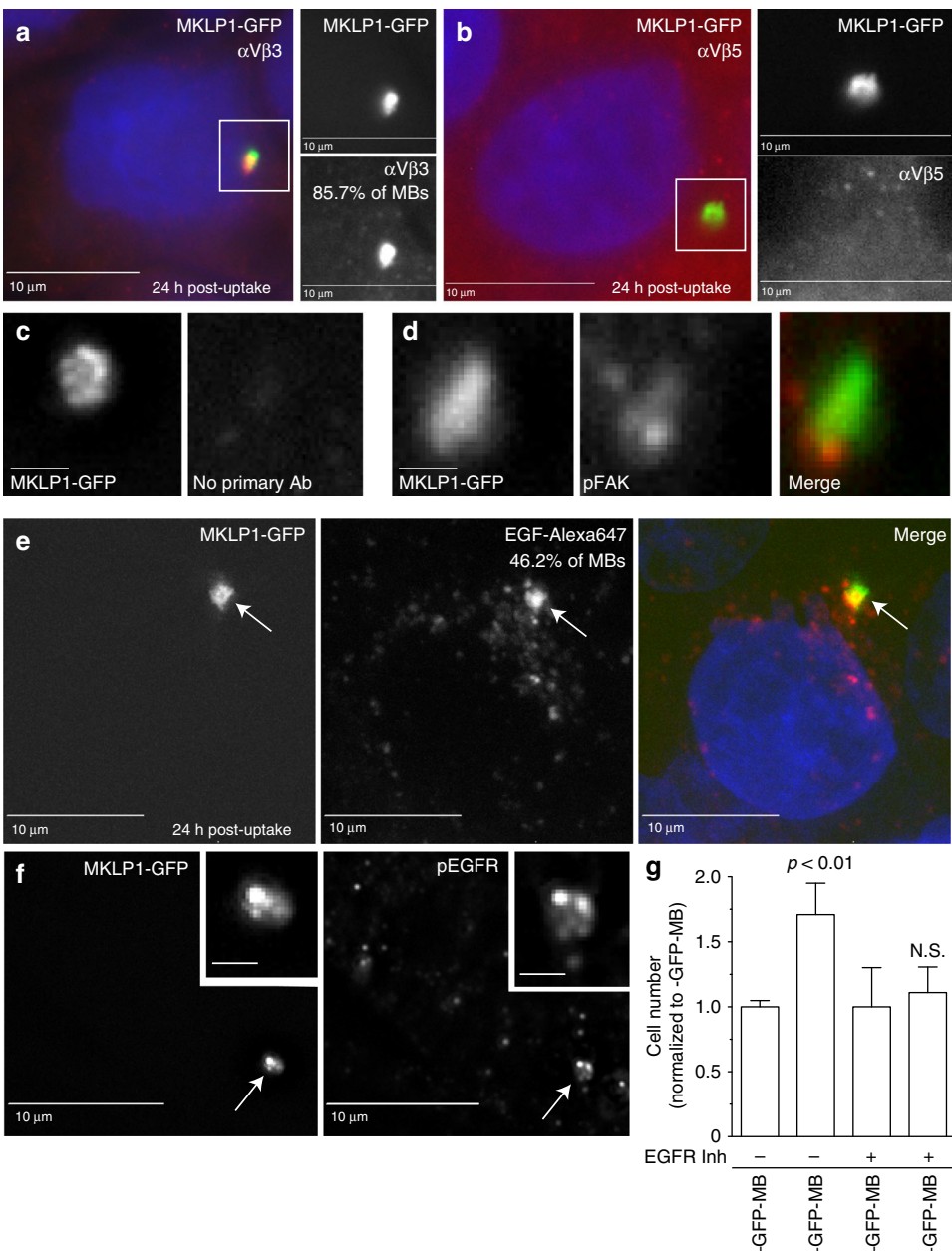

**Fig. 7** αVβ3 and EGFR mediate MBsome signaling. **a–d** HeLa cells were incubated with purified GFP-MBs for 3 h, followed by wash and another incubation for 24 h. Cells were then fixed and stained with anti-αVβ3 (**a**), anti-αVβ5 (**b**), anti-phospho-FAK (**d**) antibodies. Panels in **c** show control staining where primary antibodies were not added. Boxed regions mark the part of the image shown as a higher magnification image in the insets on the right. Scale bar is equivalent to 1 μm. **e** HeLa cells were co-incubated with purified GFP-MBs and EGF-Alexa647, followed by wash and another incubation for 24 h. Cells were then fixed and colocalization between MBs and EGF analyzed. Arrows point to MBsomes. **f** HeLa cells were co-incubated with purified GFP-MBs and non-labeled EGF, followed by wash and another incubation for 24 h. Cells were then fixed and stained with anti-phospho-EGFR antibodies. Arrows point to MBsomes. Scale bars in insets are equivalent to 2 μm. **g** HeLa cells were incubated with purified GFP-MBs. Cells were then washed and flow sorted to separate fractions with or without internalized GF-MBs. Equal number of cells from each fraction were then plated and incubated for 48 h in the presence or absence of 10 μm of EGFR inhibitor (erlotinib). Cells were then washed again and incubated for another 48 h followed by cell counting to determine the number of cells. Data shown are the means and standard deviations derived from three independent experiments (one-way ANOVA)

Another possibility is that MBsomes may signal via receptor tyrosine kinases (RTKs). Integrins have been shown to cluster with specific RTKs, particularly EGFR[38,39]. It is possible that upon internalization into MBsomes these αvβ3 integrin and EGFR clusters could remain activated and signal to affect cell proliferation. To test this, we co-incubated HeLa cells with fluorescently labeled EGF (EGF-Alexa647) and GFP-MBs. Cells were then washed and incubated for an additional 24 h to allow for MBsome formation. 46.2% of MBsomes still contained

EGF-Alexa647 after 24 h, suggesting that MBsomes may contain active EGF-EGFR complexes (Fig. 7e). Consistent with this idea, MBsomes also colocalized with activated phospho-EGFR (Fig. 7f) and neither pEGFR nor pFAK could be detected in purified post-mitotic MBs (Supplementary Fig. 4C). Finally, inhibition of EGFR using erlotinib completely blocked MB-induced increase in cell proliferation (Fig. 7g). Importantly, our RNAseq analysis shows that mRNAs of Cyclin B and Cyclin D are up-regulated in cells containing internalized GFP-MBs (Supplementary Data 4, 5B)

and both cyclins are known to be up-regulated during EGFR-induced proliferation. Taken together, these data suggest that RTK/integrin clustering and intracellular activation could be one of the MBsome-signaling mechanisms, and that the MBsome is a organelle capable of regulating a variety of cellular processes including proliferation and anchorage-dependent growth and survival.

## Discussion

Until recently, the MB was thought to be lost immediately following cytokinesis, either through release into the extracellular space or autophagic/phagocytic degradation by one of the daughter cells. However, recent studies have shown that MBs can be retained by cells after completion of cell division and that post-abscission MBs have additional functions beyond mediating cell abscission, including the promotion of stemness or tumorigenicity[7,8,11,19]. Consistent with this idea, it has been shown that MBs accumulate in stem cells and aggressive cancer cells, although direct a causal relationship between MB accumulation and stemness or tumorigenicity is yet to be demonstrated and remains quite controversial in the field. This controversy, at least in part, is a result of the use of autophagy inhibitors to promote MB accumulation, thus making it difficult to separate cellular effects caused by MBs from the ones resulting from inhibition of autophagy. In this study, we performed an analysis of the functional consequences of internalizing post-abscission MBs. We used a FACS approach to isolate cells that accumulated GFP-MBs and to analyze the changes in their transcriptome. Importantly, these analyses have demonstrated that post-abscission MB internalization leads to an increase in mRNA transcripts that enhance cell proliferation and that increased MB internalization directly stimulates cell proliferation and anchorage-independent cell survival and growth.This is the first study that demonstrates a direct causal relationship between post-abscission MB signaling and cell division as well as tumorigenicity.

An obvious caveat to using our flow sorting approach to enrich for cells containing post-abscission MBs is the possibility that it selected for cells that are naturally more proliferative and that MB accumulation is simply a byproduct rather than the cause of this increase in cell division. To address this concern, we developed a new technique designed to purify post-abscission MBs[40]. Using this technique, we could plate isolated MBs directly onto different cell populations and examine their effects. Remarkably, these isolated post-abscission MBs can be internalized by interphase cells and are retained in the cytoplasm by evading degradation for up to 48 h (Fig. 1c, d). We also have shown these MBs can then induce cell proliferation as well as enhance anchorage-independent survival/growth, the key features of increased tumorigenicity. While in this study we did not find evidence that MBs regulate cell stemness, we cannot rule out this possibility for other cell types and further work is necessary to potentially define a role for MBs in stem cells.

We show that internalization of post-mitotic MBs leads to an increase in proliferation in HeLa as well as MDA-MB-31 cell lines, supportive of the idea that post-abscission MB signaling is not just a Hela cell phenomenon but likely functions in other cell types as well. Consistently, recently published work from Lujan and colleagues demonstrated that post-abscission MBs can also regulate epithelial cell polarity in MDCK cells[41]. Based on all of these findings we propose that internalized MBs are long-lived signaling structures which affect several cellular structures and refer to them as MB-containing endosomes or MBsomes (Fig. 8).

It was previously proposed that post-abscission MBs accumulate as a consequence of asymmetric abscission, resulting in one

daughter cell inheriting the MB (Fig. 1b)[6,7]. Importantly, internalized MBs are different from the post-abscission MBs inherited after cell division, since they are surrounded by a plasma membrane remaining from the division of the mother cell (Fig. 1b). The presence of this membranous envelope suggests that internalized MBs may signal and function differently than the MBs inherited from asymmetric cell division. What these differences are and why some cells inherit MBs while others internalize MBs remains unclear and will require further investigation. In this study, we focused on identifying machinery and functional consequences of MB internalization. Previous work has shown that in HeLa cells MBs can be internalized and degraded either by a neighboring cell or by one of the daughter cells[6]. Here we show that during interphase, several cell types, namely HeLa, MDCK, and MDA-MB-231 cells, can internalize extracellular MBs via an actin-dependent phagocytosis-like mechanism. This supports the idea that post-abscission MBs may contain a signal that mediates their recognition and stimulates MB internalization. Consistent with this hypothesis, we demonstrate that PS is enriched at the outer leaflet of the MB envelope and mediates MB engulfment by binding to αVβ3/αVβ5 integrin and matricellular receptor complex (Fig. 4d). Intriguingly, PS is not present at the outer leaflet of the MB plasma membrane during cell division and only becomes exposed after MBs are released to the extracellular environment, suggesting that MBs probably need to 'mature' to become internalization-competent. Requirement for such a maturation step would allow MBs to diffuse away from the dividing cell and be internalized by neighboring cells, thereby mediating lateral information transfer in a manner like exosomes. Further work will be needed to characterize the regulation of MB maturation and internalization.

Typically, phagocytosed structures are rapidly degraded within 2–5 h by fusing with lysosomes. We observed that MBsomes can evade degradation for up to 48 h post-feeding, potentially allowing for continuous signaling. Certain species of bacteria are capable of hijacking host cell actin polymerization machinery, allowing for the formation of the protective actin patches and delaying lysosomal fusion[25,42]. These actin patches are usually very dynamic, can drive bacterial propulsion, and can regulate lysosomal targeting[36,37]. Similarly, we show that MBsomes are associated with dynamic actin patches that can mediate intercellular MB movement and appear to delay MBsome degradation. Consistent with this hypothesis, actin depolymerization decreased the number of MBsomes and led to an inhibition in MBsome-dependent cell proliferation. We hypothesize that MBsome-associated actin patches suppress MBsome degradation and increase signaling that promotes cell proliferation (Fig. 8).

Our last remaining question is how the MBsome signals to increase proliferation. Our data suggest that internalized MBs are surrounded by a double membrane, although it is possible that eventually this membrane may undergo back-fusion, releasing the MB into the cytosol. Despite this possibility, the MBsome needs to transmit signal through both the MB envelope and the phagocytic membrane, and this signaling likely involves transmembrane receptors. We found that accumulation and internalization of MBs leads to an increase in anchorage-independent survival and growth, a process that is dependent on integrin activation and signaling[43,44]. We speculated that the MBsome may signal through an integrin-dependent pathway. We show that extracellular MBs are recognized and internalized, at least in part, through αvβ3/αvβ5 integrins. We also demonstrate that even 24 h after internalization the MBsome still contains activated αvβ3 integrin complexes (Fig. 7). This raises the possibility that the MBsome-integrin signaling cascade may continuously activate proteins downstream of cell adhesion pathways, such as FAK and Src. In these instances, the cell could continue to proliferate even

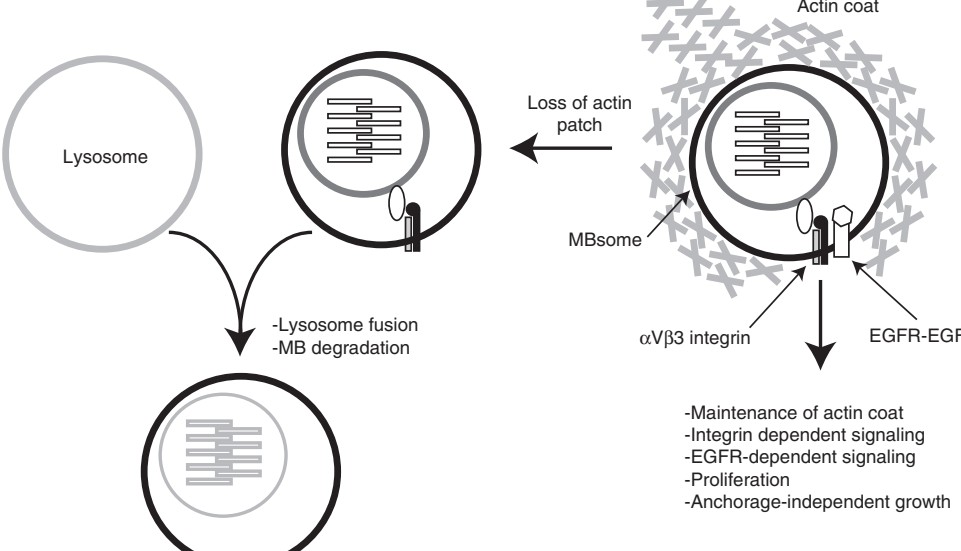

**Fig. 8** Schematic representation of the proposed MBsome regulation and function

in the absence of adhesion to the extracellular matrices (inside-in signaling). Another possibility could involve RTK. Previously published work suggests that integrins and RTKs, especially EGFR, can cluster on the cell surface, allowing them to cooperate for downstream signaling[38,39]. Other work has indicated that EGFR is only fully functional once it has been internalized and continues signaling from specialized endocytic compartments known as signaling endosomes[45]. Consistent with this hypothesis, about 50% of MBsomes contain activated EGF-EGFR signaling complexes, suggesting that MBsomes can signal via at least two different pathways: αVβ3-FAK-Src and EGF-EGFR (Fig. 7g).

In conclusion, we demonstrate that PS-αVβ3/αVβ5 integrin-mediated engulfment of post-mitotic MBs results in the formation of MBsomes, a signaling organelle that regulates cell proliferation, as well as anchorage-independent survival and growth. We also demonstrate that MBsome-associated actin coats allow MBsomes to evade lysosomal degradation and to mediate long-term signaling via EGFR and αVβ3-FAK-Src signaling pathways. These findings identify a unique mechanism for regulating cell proliferation and cancer cell tumorigenicity.

## Methods

**Cell culture and treatments**. Cells were kept in 37 °C humidified incubator at 5% $CO_2$, routinely tested for mycoplasma, and were maintained in DMEM with 5% FBS and 1% penicillin/streptomycin. To create the mCherry-CAAX stable cell line, HeLa cells were infected with the lentivirus pLVX:mCherry-CAAX. The population was selected with puromycin, and stable clones were isolated. To create the LifeAct-mCherry stable cell line, HeLa cells were infected with retrovirus (Addgene #54491, Michael Davidson). The population was selected with puromycin, and stable clones were isolated. For cilengitide treatments, cells were incubated with 40 μM cilengitide, an RGE control peptide or left untreated for 30 min at 37 °C. Cells were then added to collagen-coated coverslips and allowed to adhere. Thymidine–nocodazole double synchronization was performed as previously described[3]. Briefly, cells were treated with 5 mM thymidine overnight, allowed to recover for 8 h, treated with 0.1 μg/ml nocodazole overnight, and washed. For latrunculin treatments, cells were treated with Latrunculin A for 30 min at a final concentration of 0.5 μM, washed, then allowed to recover for 8 or 72 h. For cytochalasin D and Rac1 inhibitor treatments, cells were treated with either 10 μM cytochalasin D or 100 μM Rac1 inhibitor (NSC23766). For erlotinib treatments, cells were allowed to adhere post-sorting, then treated at a final concentration of 10 μM for 3 days. Erlotinib was then removed, and cells were counted at 3 and 5 days post-sort.

**Flow cytometry and cell sorting**. For cell sorting prior to RNAseq, MKLP1-GFP cells were sorted into low GFP expressing and high GFP-expressing populations using a Beckman Coulter MoFlo XDP100. For cell sorting following MB feeding, cells were fed midbodies, washed 3 h post feed, then sorted into GFP− and GFP+ populations 24 h post-feed using a MoFlo XDP100. For cell cycle analysis, MKLP1-GFP cells were first sorted into low GFP or high GFP-expressing populations using a MoFlo XDP100. Cells were then resuspended in Krishan's propidium iodide stain, incubated overnight, and DNA content was assessed using a Beckman Coulter Gallios.

**RNA isolation and quantitative PCR**. RNA was extracted using Trizol per the manufacturer's instructions. RNA was reverse transcribed using iScript (Bio-Rad). Quantitative PCR was performed using SYBR Green (Bio-Rad). Primer sequences can be found in Supplementary Data 3.

**Antibodies and plasmids**. The following antibodies for immunofluorescence were used: acetylated tubulin (Sigma, T7451, 1:100), CD63 (gift from Dr. Andrew Peden, 1:100), Ki67 (ThermoScientific Clone SP6), αVβ3 (R&D MAB3050, 1:100) and αVβ5 (R&D MAB2528, 1:100) (R&D Systems), αVβ3 (Abcam Ab190147, 1:100), MFGE8 (Santa Cruz sc-8029, 1:100), pFAK (Tyr397, Invitrogen 44624G, 1:100) and pEGFR (Y1068, Cell Signal #3777, 1:100). The following antibodies for western blots were used: Cep55 (Abnova H00055165, 1:500), beta-tubulin (LiCor 926-42210, 1:1000), PDI (Cell Signaling C81H6, 1:500), MFGE8 (Santa Cruz sc-8029, 1:1000), and alpha-tubulin (Santa Cruz 23948, 1:1000 Original and uncropped AlexaFluor-594- and AlexaFluor-488-conjugated anti-rabbit and anti-mouse secondary antibodies were purchased from Jackson ImmunoResearch Laboratories (West Grove, PA). AlexaFluor-568-phalloidin was purchased from Life Technologies (Carlsbad, CA). Phalloidin iFluor488 was purchased from Abcam (Cambridge, UK). The IRDye 680RD Donkey anti-mouse and IRDye 800CW donkey anti-rabbit secondary antibodies used for western blotting were purchased from Li-COR (Lincoln, NE). To generate pLVX:mCherry-CAAX, mCherry-CAAX was subcloned from p3E-2A-mCherry-CAAX and inserted into pLVX. To generate dominant-negative MFGE-8, MFGE-8-C2 was subcloned from mRFP-LactC2 (Addgene #74061, Sergio Grinstein) and inserted into pGEX-KG. Recombinant GST-tagged protein was eluted with 25 mM glutathione, dialyzed, and used for subsequent assays. LifeAct-mCherry was obtained from Addgene (#54491, Michael Davidson).

**Lipid-binding assay**. Liposomes were generated using previously described protocols, according to Avanti Polar Lipids. Briefly, different ratios of PC and PS were dried under nitrogen gas, resuspended in HEPES-buffered saline, and sonicated. For binding assay, 5 μg of recombinant MFGE-8-C2 was added to a no lipid control, PC only (1 mg), or different PC:PS ratios (2:98, 20:80), incubated for 30 min, and pelleted for 20 min at 50,000 RPM using a TLA-55. Supernatants and pellets were separated into different tubes, run on SDS–PAGE gels, and stained with Coomassie.

**MB purification, feeding, and treatments**. Midbodies were isolated as previously described (Peterman and Prekeris, 2016). Briefly, media from MKLP1-GFP cells was collected and subjected to a series of centrifugation spins (300 × *g*, 10,000 × *g*). Sucrose gradient fractionation was performed at 3000×*g*, and the

interphase between 40% glycerol and 2 M sucrose was collected, spun at 10,000 × g, and resuspended in PBS. Midbodies were plated onto cells. For all feeding experiments about 90,000 isolated MBs were added to 200,000 cells plated in 0.5 mls of media (ratio of 0.45 MBs per cell). Cells were washed 3 h post-feeding if the assay being performed was longer than 3 h. For mCherry-MKLP1 midbodies isolated from 293T cells, cells were first electroporated with mCherry-MKLP1 (Addgene #70154), allowed to recover for 3 days, then MBs were isolated as described above. For GST-PS-BD assays, midbodies were resuspended in PBS and recombinant GST-PS-BD or GST was added at a final concentration of 0.08 mg/ml. For EGF-labeling assays, EGF-Alexa647 (Invitrogen Inc.) was mixed with MBs at a final concentration of 20 µg/ml and incubated with cells for 3 h. Cells were then washed and incubated for 24 h before fixing and immuno-fluorescence analysis.

**Immunofluorescence and time-lapse microscopy.** Fixed cells were imaged with an inverted Axiovert 200M microscope (Zeiss) with a × 63 oil immersion lens and QE charge-coupled device camera (Sensicam). Z-stack images were taken at a step size of 500–1000 nm. Image processing was performed using 3D rendering and exploration software Slidebook 5.0 (Intelligent Imaging Innovations). Images were not deconvolved unless otherwise stated in the figure legend. For time-lapse imaging cells were plated on collagen-coated glass bottom dishes. Time between images is stated in the figure legends of each time-lapse. For 3D structured illumination microscopy (SIM), cells were imaged using a Nikon nSIM Super-resolution Microscope System.

**Electron microscopy and tomography.** Cells were chemically fixed in 2% glu-taraldehyde, followed by post-fixation in 2% osmium tetroxide, ethanol dehydration, and embedding in Embed812/Araldite (Epon) resin. Microtomy of embedded cells was performed on a Leica UC6 Ultramicrotome for 90 nm thin sections, made at the beginning and end of each block, with 250 nm semi-thick sections making up the bulk and near-entirety of the serial sections. Thin sections were used for quality control and semi-thick sections were used for tomography. Semi-thick sections were collected on formvar-coated rhodium-plated copper slot grids and labeled with 15 nm fiducial gold. Thin sections were first mapped on a ThermoFisher (Phillips) CM10 80 kV TEM. Dual-axis tilt series of semi-thick sections were col-lected from ±60° with 1° increments at 200 kV using a ThermoFisher (FEI) Tecnai F20 FEG-TEM, and with a Gatan US4000 4k CCD, and SerialEM software[46] for tomography. Tilt series of cell 2A at a magnification of ×9600 (2.266 nm pixel size) and cell 2B at ×11,500 (1.928 nm pixel size). Average tomographic resolution, based upon thickness and tilt was ~6 nm. IMOD software was used to construct tomograms using weighted back-projection[47]. 3dmod software was used for the manual segmentation of membranes and model measurements. 3D, meshed structures were generated from manually assigned contours using periodic mod-eling of closed contours.

**Anchorage-independent growth soft agar assay.** Soft agar assays were per-formed as previously described[48]. Briefly, an agar solution was overlaid on six-well dishes as a base layer. 200–500 cells were resuspended in agar solution and overlaid on top of the base layer. Phenol red-free and serum-supplemented media was overlaid on top of cells once agar layer had solidified. Colonies were stained with NBT 2 weeks following plating, imaged, and quantified with Metamorph.

**RNAseq library preparation and sequencing.** For final RNA preparations, the RNA was suspended in RNase-free water and its RNA purity and concentration were measured on an Agilent Bioanalyzer (Agilent Technologies, Palo Alto, CA). A total of 200–500 ng of total RNA was used to prepare the Illumina HiSeq libraries according to manufacturer's instructions for the TruSeq RNA kit. Three different libraries were generated for every condition (GFP− and GFP+) from independent cell sortings. The mRNA-seq libraries were sequenced using next-generation sequencing technology on the Illumina HiSeq2000 platform at the University of Colorado's Genomics and Microarray Core Facility, utilizing 1 lanes, single read 125 cycles. For the RNA-sequencing preparation from fed cells; cells were fed MBs, incubated for 24 h, and sorted into GFP− and GFP+ populations for subsequent RNA isolation and library preparation.

**Bioinformatic analysis of transcriptome.** FASTQ files were then mapped to Human Genome Sequence using the Tophat aligner algorithm (Trapnell, Pachter and Salzberg, 2009) under the single-end read setting. The count table was generated using Bioconductor package; GenomicAlignments[49]. The resulting SAM/BAM files were processed, filtered, and eventually perform differential expression analysis using the Ritchie al.et algorithm[50]. We used statistical programming packages from the Bio-conductor consortium (www.bioconductor.org) for downstream data analysis, annotation, and visualization. Briefly, count data were filtered to remove genes with zero count. The data were further normalized with the "edgeR" package, followed by the "limma" package with its "voom" method, linear modeling, and empirical Bayes moderation to assess differential expression. Genes annotation were performed using the "AnnotationDbi" package. The principal component analysis (PCA) visualization was performed using the "ggplot2" package. The heatmaps were generated using the "Heatplus" package. For annotation cluster analysis, all genes were assessed by the DAVID Bioinformatics tool for clustering.

**Proteomic analysis of purified midbodies.** The protein samples of MBs were subjected to filter-aided sample preparation (FASP) trypsin digestion as described[51]. Mass spectrometry analyses were performed as previously described[52]. Briefly, the peptide mixtures were injected onto reversed-phase trap column and subsequently separated on HSS-T3 C18 75 µm × 250 mm analytical column (Waters Corporation, UK) in 90 min linear gradient. Data were acquired in positive ion mode with Synapt G2 mass spectrometer (Waters Corporation, UK). LC–MS data were collected using data-independent acquisition (DIA) mode MS$^E$ in combination with ion mobility separation. Raw data files were processed and searched using ProteinLynx Global SERVER (PLGS) version 2.5.3 (Waters Corporation, UK). The following parameters were used to generate peak lists: minimum intensity for precursors was set to 150 counts, minimum intensity for fragment ions was set to 50 counts and intensity was set to 500 counts. Minimum identification criteria included 1 fragment ions per peptide, 3 fragment ions per protein, and minimum of 2 peptides per protein. UniprotKB/SwissProt human database (2015-04-22) was used for protein identification.

**Cell proliferation assays.** For Ki67 intensity analysis, random fields of GFP-MB fed cells were imaged. In each experiment, the same exposure for different fields was used to ensure continuity. For mitotic index quantification, random fields of GFP-MB fed cells were imaged.

To assess cell proliferation, two different techniques were used. For Fig. 2c, HeLa mCherry-CAAX cells were plated at a low density into glass bottom dishes. MBs were purified and fed onto these cells, leaving one dish as an unfed control. Cells that had begun to engulf MBs after 3 h (as indicated by mCherry-CAAX) were then imaged and the position of the cell was recorded. Colony size was reimaged 7 days post feeding. To assess proliferation of MDA-MB-231 cells (Fig. 2f) and HeLa cells treated with latrunculin A (Fig. 6f), cells were fed MBs and allowed to internalize them (~24 h). Cells were sorted into −GFP-MB and +GFP-MB populations via flow cytometry and seeded onto 24-well plates that had been divided into four equal quadrants. Total cell number in each quadrant was counted after seeding, then tracked and counted for 3 days.

**Correlative light and electron microscopy.** HeLa cells were grown on Cryo-Capsules[53] and fed with MBs after 24 h. The day after, cells were high-pressure frozen using an HPM-Live µ (CryoCapCell) and freeze-substituted for 12 h with 0.01% uranyl acetate/0.05% glutaraldehyde/1% H$_2$O in anhydrous acetone[53]. Cells were embedded in Lowicryl HM20 and bloc face were imaged for fluor-escence of the MBs. Cells presenting MB fluorescent profiles were mapped according to the carbon landmarks of the CryoCapsule for targeted microtomy. Ultrathin sections of 70 nm were collected on slot grids, post-stained using uranyl-acetate (4% in H$_2$O), and Reynolds lead citrate for 5 min each. Sections were then imaged at the electron microscope (Tecnai Spirit, Thermo Scientific), and identified cells were relocalized using eC-CLEM software (pmid: 28139674) on Icy[54].

**Statistical analysis.** All statistical analyses were performed using GraphPad Prism Software (GraphPad, San Diego, CA). A Student's t-test was used to determine significance unless otherwise noted. Error bars represent standard deviation unless otherwise noted. For all immunofluorescence experiments, at least five to ten randomly chosen image fields were used for data collection. For quantitative immunofluorescence analysis, the same exposure was used for all images in that experiment and was quantified using Intelligent Imaging Innovations software (Denver, CO, USA).

## Data availability

All data are available from the authors upon request. The mass spectrometry proteomics data have been deposited to the ProteomeXchange Consortium via the MassIVE partner with the dataset identifier MSV000083826 and RNAsequencing have been deposited to GEO with Accessioncode GSE131662.

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

## Acknowledgements

We would like to thank Drs. Chad Pearson (UC AMC), Jeffrey Moore (UC AMC), Ahna Skop (University of Wisconsin), and Lai Kuan Dionne (Washington University of St. Louis) for critical reading of the manuscript. We are also grateful to Dr. Andrew Peden (University of Sheffield) for anti-CD63 antibody, Dr. Steven Doxsey (University of Massachusetts Medical School) for MKLP1-GFP-expressing HeLa cells, and Drs. Lynn Heasley and Traci Lyons for various reagents. We would like to acknowledge the University of Colorado Cancer Center Flow Cytometry Core Facility, the University of Colorado Genomics and Microarray Core Facility, and Garry Morgan and Matthew West at the University of Colorado Boulder Electron Microscopy Service. This work was supported in part by a grant from NIH-NIDDK (DK064380 to R.P.), grant from Research Council of Lithuania (APP7/2016 to V.A.S.) and Mayent-Rothschild-Institut Curie Award (to R.P.). P.G. is supported by WFS National Scholarship Program. This work was also supported by the French National Research Agency through the "Investments for the Future" program (France-BioImaging, ANR-10-INSB-04). We also acknowledge the PICT-IBiSA, member of the France-BioImaging national research infrastructure, supported by the CelTisPhyBio Labex (N_ ANR-10-LBX-0038) part of the IDEX PSL (N_ ANR-10-IDEX-0001–02 PSL).

## Author contributions

E.P. and R.P. conceived the project and wrote the manuscript, with the help of P.G. and J.S. Proteomics was performed by P.G., V.A.S., A.K., and M.V. Correlative light and electron microscopy of internalized midbodies was performed by X.H., I.H., and G.R. The remainder of the experiments were designed and performed by E.P. and R.P.

## Additional information

**Competing interests:** The authors declare no competing interests.

