## [Peer Review File · Nature Communications]

Reviewers' comments:

Reviewer #1 (Remarks to the Author):

The study by Peterman et al identified the mechanisms that regulate post-mitotic MB retention as well as consequences of MB accumulation. To this end, the authors used RNA-seq to demonstrate that accumulation of post-mitotic MBs leads to an increase in transcription of genes associated with proliferation. In order to ask whether RNA-seq reflects the functional cause of MB accumulation, the authors developed unique technique to isolate extracellular MBs. By using this, the authors showed that interphase cells can uptake post-mitotic MBs and that accumulation of these MBs leads to increased proliferation and anchorage independent growth and survival. Mechanistically, the recognition and internalization of post-mitotic MBs was found to be dependent on phosphatidylserine binding that is similar to phagocytic process. The author also demonstrate that internalized MBs form membrane-bound organelles (denoted MBsomes) and that these MBsomes are protected from lysosomal degradation by the formation of dynamic actin coats. Finally, the author found that internalized MBsomes trigger EGF receptors (EGFR) and $\alpha V\beta 3$ integrin pathways to increase proliferation.

Overall, the approach is impressive and interesting, especially the use of isolated MB to address the functional consequence of MB uptake as well as the mechanisms of internalization which has not been fully addressed in the field. There are multiple interesting observations here, but the following issues need to be considered to justify the claims by the authors and the quality of the study.

Major concerns

Others have shown that MBs can be internalized (i.e. Crowell et al, 2014, showed loss of MB when entering an acid compartment, and Pohl and Jentsch, 2009, showed colocalization of MB with autophagy components). The data on MB internalization is not particularly convincing. The only data that suggests internalization is the colocalization of MB with CD63, but if MB are in the lysosome how do the authors reconcile that they are still GFP positive since the GFP signal is quickly lost in acidic lysosomes (Fig. 1C, D)?

Other concern is the preparation of MB. Does it contain exosomes or microvesicles? Other contaminants in these preparations could be responsible for the effect on proliferation. Moreover, the control is "unfed" cells. Is it possible to isolate MB that are not functional (mutant of MKLP1 or others, "empty liposomes?") to use as controls? Also, in the physiological setting, how many MBs do cells usually take up and what percentage of cells internalize MBs? Is it possible that exogenously added MBs are not reflecting what is occurring in the natural culture condition?

Other concern is that authors don't describe what statistical tests are used to analyze the data (and in some figures [Fig. 1D, 2B, 2C, Supl. Fig.1E], there is no statistical analysis).

Authors show the effect of MB on HeLa cells and other cell types on cell proliferation, but it would be useful to also show this in other cell types.

Specific Figure related comments:

Figure 1 (and Suppl. Fig. 1)

Authors do not provide enough convincing data to show that MB are internalized and not just "bound".

Is it possible to label MB with pHRodo or Cypher5E to definitively show that they are internalized and reach the lysosomal compartment?

Authors argue that MB evade lysosomal degradation because a fraction of MBs is not associated with CD63. This seems to be an overstatement and needs to be shown convincingly.

Suppl. Fig. 1B shows different GFP intensity: which fraction was sorted?

Suppl. Fig. 1C shows that -GFP-MB cells still have MB (~ half of +GFP+MB). This is a concern when no (or little) differences are detected in gene expression of stemness markers (or Ki67 in Suppl. Fig. 1E; is this significantly different?)

Figure 2

Authors claim that MB induce cell proliferation and anchor-independent growth. As stated above, the concern here is the MB preparation, and what are the contaminants? Is there a better control than "unfed" cells?

Does PS-BD or RGD (or annexin V) inhibit the effect on proliferation?

Fig. 2A shows that the effect on proliferation (24h) is only observed on cells with >2 MB. This should also be taken into consideration in Fig. 2E (7d) where 2 populations are observed (+GFP+MB group): one with low number/colony and another with high number/colony. Is it possible that the increased proliferation is a product of more cargo, more cellular content, change in cell volume? Will artificial beads stimulate proliferation? Again, a better control is necessary (not just "unfed" cells).

Why do the authors show different time points for different cells? Are there differences in the kinetics between cell types – this needs to be explained. Also, MDA-MB-231 cells don't show an actin coat (Suppl. Fig. 2D) as HeLa cells (Fig. 6A). This needs to be reconciled with the effect on proliferation?

Fig. 2D: Are these the FACS sorted cells used in experiments in Fig. 2F? Do the cells conserve the MB after 14 days of culture?

Figure 3

The authors test whether the MB are "internalized" via fusion or engulfment and show that MB are surrounded by membrane protrusions. These data doesn't show either fusion or engulfment nor internalization. Also, there are membrane protrusions where no MB are found. What is the percentage of MB associated to protrusions?

Again, if most MB are associated with the lysosomal marker CD63, why GFP is not quenched by the acid pH?

If MB are targeted to the lysosomes, Cypher labeling might help to show this.

Figure 4

Authors show colocalization of actin protrusions and MB.

What is the effect of an actin inhibitor (such as EHT1864) or Cytochalasin D in the binding and "internalization" of MB?

(Lata is used in Fig. 6 after internalization to evaluate how MB evade degradation)

Figure 5

Authors show nicely that both isolated MB and "endogenous" MB (Fig. 5G) externalize PtdSer using PS-BD. Does the control GST show any non-specific binding? The staining with the control construct is not shown and the GST alone seems to have a mild effect on MB binding (Fig. 5B).

It would be useful to add another staining for PtdSer, such as Annexin V.

Authors study the effect of blocking PtdSer (PS-BD) and integrins (RGD) on the binding (3h) and internalization (24) of MB. Again, authors do not provide convincing data to show that the MB are really internalized (the different time points is not enough).

Fig. 5E: Why RGE and GST controls are not shown? Do they have any effect?

Have authors use antibodies anti $\alpha V\beta 3$ as an alternative block?

Figure 6

Here, authors evaluate a possible role of the actin “coats” to evade MB degradation. What is the fraction of MB with actin coats? Because the fraction of MB that “evade degradation” is minor. LatA “reduces” the number of MBsomes (what is the statistical test used?). Are more MB associated with CD63 after LatA treatment? A reduced number of MBsomes is not conclusive.

LatA doesn't have an effect on proliferation on either –GFP-MB or +GFP+MB cells. The 3rd and 4th bars are not different.

Effect is analyzed after 36h without LatA? Is LatA reversible?

Is this cell type specific? MDA-MB-231 cells do not show actin coats (Suppl. Fig. 2D). Please explain.

Figure 7

May be I'm not interpreting the photos correctly, but why are the integrins only detected on the MB and not on whole Hela cells? These MBs were derived from Hela cells? The co-staining with pFAK and pEGFR are correlative, they do not necessarily show a requirement for gene expression and/or MB internalization. Authors will need to use mutants or inhibitors to show that these pathway is involved. Do PS-BD or RGD block pFAK and pEGFR localization with MB?

Minor points

1) Figure 5D should say binding not uptake

4) Typos:

“my microscopy” line 618, page 14

“lipososomes” line 640, page 15

Reviewer #2 (Remarks to the Author):

This manuscript addresses the unsolved question of whether midbodies (MB) play a potential role in cell proliferation. MBs/MB derivatives are generated after cytokinetic abscission and released into the extracellular medium or internalized by phagocytosis in *C. elegans*, *Drosophila* and numerous cultured cells before being degraded by lysosomes. Proliferative cells tend to harbor more MBs and cells with high numbers of MBs are more prone to grow in soft agar. However this literature is highly controversial because it is essentially based on correlative evidence. In this manuscript, the authors purified MBs from the cell medium and analyzed the effect of adding them to cells. They report that addition of MBs increased cell proliferation and the expression of proliferative genes, and promoted cell growth in soft agar. Purified MBs were internalized by a phagocytic-like mechanism, through a phosphatidylserine/phosphatidylserine-bridging proteins/ $\alpha\text{v}\beta\text{3}$ integrin-dependent mechanism. They suggest that internalized MBs escape degradation thanks to an actin-rich coated structure. Also MBs surrounded by a phagocytic membrane (termed MBsome) are proposed to promote proliferation through EGF-receptor/ $\alpha\text{v}\beta\text{3}$ signaling pathways. The authors use the correct strategy and provide the first attempt to experimentally tackle the controversial question of whether MBs play a role in cell signaling and proliferation. However, as it stands, there are major concerns that must be addressed before solid conclusions can be reached.

Major issues

1- Many crucial experiments (e.g. Figure 2A, 2B, 2D, 2E, 2F, 2G) are based on the comparison between cells incubated with or without purified MBs. Unfortunately, there is no real characterization of the degree of purity of the added MBs, which are isolated from the extracellular medium of cultured cells. MB proteomic analysis (Suppl. Table 2) actually demonstrates that the preparation contains many contaminants while many key MB proteins are lacking (Cep55, PRC1, KIF20A...). How pure is this MB prep? Is it contaminated with cells (as suggested by the proteomic analysis), exosomes, intracellular/plasma membranes, RNAs etc? As a better negative control, the authors should treat the cells with MB-negative purifications from cells that do not produce MBs, which would include all the potential contaminants.

2- Once the authors resolve the contamination issue, the physiological relevance of the experiments should be addressed. Indeed, it seems that HeLa cells exceptionally contain more than 2 MBs per cell. One issue of the manuscript is that there is no indication of the number of MBs added per cell in order to observe a biological effect. What is the mean number (and number distribution) of exogenous MBs in Figure 2A, 2B, 2D, 2E, 2F, 2G? How does it compare to the distribution of MB numbers per cell in an untreated population? It is also very likely that the number of added GFP-positive MBs, based on GFP-positive signals, is underestimated by the authors. Since GFP is quenched in acidic compartments, antibodies against GFP should be systematically used to address the localization and number of added MBs per cell (e.g. Figure 1D, 3D). How many MBs detected with anti-GFP are found after 3h, 24h and 48h?

3- Related to this question, how solid is the evidence that MBs are internalized and not merely at the cell surface after 24h of incubation? The authors must demonstrate that the washing step indeed removes non-internalized MBs. Z-stacks should also be systematically provided when relevant (Figure 3, Figure 6, Figure 7). For instance, the MBs in Figure 6A, 6C, 7A seem to be located at the cell surface (with an actin phagocytic cup described in *C. elegans* and in cultured cells), which is not consistent with MBs in MBsomes.

4- Once the contamination issue is resolved, a RNAseq experiment should be carried out on cells treated with control vs. MB+ purifications. This would provide a global characterization of the variation in gene expression upon MB treatment and would be much more informative than the experiment presented in Suppl. Table 1.

5- The notion of signaling by the MBsome (Figure 7) is interesting but it is not sufficiently supported by the data.

First, the authors use anti- $\beta 3$ integrin and anti-EGFR antibodies to suggest that MBsomes are signaling vacuoles. However, EGFR was already found in the proteome of intact MBs (suppl. Table 2). Therefore, are we looking at EGFR present at the plasma membrane of MBs or at the surface of the MBsome? The authors should demonstrate that EGFR/pEGFR staining (and potentially $\beta 3$ integrin staining) is not detected on intact MBs or on MBs at the cell surface. The authors suggest that CD63- MBs escaped degradation and function as MBsomes that could thus signal. If 87% of MBs are in $\alpha v\beta 3$ -positive MBsomes (Figure 7A), most MBs should be in CD63-negative compartments, which apparently contradicts the results presented in Figure 1D (only 30% of the MBs are CD63-). Why is there such a difference? The same holds true for the EGFR. Again, these results might be more consistent with the idea that many MBs are not internalized even after 24h and that we are looking at the EGFR/ $\beta 3$ -integrin of the MB plasma membrane. Ideally, correlative EM should be provided (showing that the EGFR or integrins are activated at the surface of MBsomes).

Second, it is crucial to demonstrate that the effect of MBs on cell proliferation is exerted via signaling from $\beta 3$ integrin or EGFR. To this aim, the RNAseq experiment requested in the previous

point should be compared with $\beta 3$ integrin-depleted or EGFR-inhibited cells.

Specific issues

1- Figure 3D: the absolute number of MBs (see also Major concerns #2) should be provided, not the % Total. How many exogenous MBs per cell are CD63+/CD63- after 3h, 24h and 48h? How many are eventually degraded? How efficient is MB internalization?

2- Figure 4 is incremental. It is already known in the literature that MBs are internalized by a phagocytic-like mechanism relying on F-actin in mammalian cells and in *C. elegans* (LC3-dependent, Rac1-dependent phagocytosis). The abstract should be revised accordingly. This Figure would be more informative if the authors could show that this phagocytic event is also Rac1-dependent in mammalian cells. The localization of CCN1, EDIL3 and/or MFG-E8 around MBs should also be provided.

3- Figure 6: the notion of the actin "coat" is not well established. The only clear coat is seen in Figure 6A, but I believe that this is rather a phagocytic event. What is the percentage of MBs displaying this actin staining? In Figure 6C, there is no visible coat. Why would it not be present here like in Figure 6A?

4- Figure 5E: The GST alone and RGE alone controls should be included. In addition, the number of internalized MBs detected with anti-GFP would be very informative.

5- Figure 6D: why is this an "MBsome"? Is it CD63-?

Figure 6F: are the conditions +GFP with DMSO vs. LatA statistically different?

LatA treatment is known to inhibit MB phagocytosis. Could it be that the observed effect is the consequence of decreased MB internalization (if a fraction of MBs are still at the cell surface after 24h, see major issue #3).

In addition, does the addition of LatA promote the recruitment of CD63/Lamp1 around MBs and their subsequent degradation?

6- Figure 2A: pictures should be provided. Does the Ki67 staining scale with the number of exogenous MBs per cell? What is the average number of "2+ GFP-MBs"?

7- Figure 2B: FACS analysis of the cell cycle (including M phase) should be provided. Are the - vs. + GFP-MBs statistically significant?

8- Figure 2D: Error bars and statistics should be provided.

9- Lines 69-70 page 3: cite the relevant literature in *C. elegans* and mammalian cells.

Lines 182-184 page 5: this has already been addressed in *C. elegans* and mammalian cells.

Lines 206-208 page 6: this has already been addressed in *C. elegans* and mammalian cells.

Confirmation is not a problem but previous literature should be cited.

10- Figure 6D: the correlation is not obvious. Please present single channels and arrows in the snapshot.

11- In the title, "post-abscission midbody" or "post-cytokinetic midbody" would be more appropriate.

Reviewer #3 (Remarks to the Author):

The present manuscript from Peterman and colleagues characterizes the role of MBsomes in signalling and cell proliferation. The manuscript is well written and provides interesting new insights into a cellular structure that is still poorly characterized.

One of the claims of the present study is that the internalization of midbodies by cells, through phagocytosis, leads to the formation of an organelle referred to as MBsomes by the authors, that can then actively send signals to influence cell functions and proliferation. A key control that is lacking for this demonstration is to rule out the possibility that the X Y Z are simply induced by triggering phagocytosis. The authors should perform experiments to test whether the internalization of inert particles (e.g. latex beads or zymosan) has the same effect on the transcription of genes promoting cell division (they might have to be opsonized to be internalized by HeLa cells).

Furthermore, how these post-mitotic MBs signal and how they evade cellular degradative machinery remains essentially unknown. Thus, we set out to determine the function of post mitotic MBs and how/if they signal to affect cellular functions.

Fig. 1C: The labeling for CD63 in the lower panel cannot be presented as representing a membrane labeling around a MB. In such cases, one should see a narrow rim around MB. What is shown here looks more like blobs. To address this important point in a more convincing way, the authors should express a fluorescently-tagged version of CD63 and follow phagocytosis in these cells.

Fig. 1E does not present statistics. Is that because the measurement was performed only once? If so, it should be repeated and statistics presented. Some negative controls should also be included.

Fig. 2A: Images of the expressing cells should be shown. 2B, C and D: Why is there no error bars and/or statistics?

Fig. 3A and B: The legend indicates an incubation of 3h. What is the "1h" marking on the image refers to?

Fig. 3C: I am not convinced by the image stating that a MB is surrounded by a membrane (arrow). Such claim would be more substantiated by EM analyses.

Fig. 3E: The images are of very poor quality and the CD63 labeling is not convincing. Idem for Fig. 3F (super pixilated). Furthermore, in Fig. 3F, I find the claim that MBsomes have 2 membranes inconclusive. Again, EM analyses would potentially solve this issue.

Fig. 4: This figure is more like a parenthesis in the study to show that the phagocytosis of MBs involves actin. The requirement of actin in all form of phagocytosis is well known and documented.

Fig. 6A-D: This part of the manuscript describes the association of MBsomes to actin structures (i.e. coats and tails). Fig. 6E: The authors observe a decrease in the number of MBsomes upon treatment with Lantraculin (an actin depolymerizer), concluding that the decrease is due to degradation. Although this could be the case, what they simply observe is a decrease in MBsomes number. To claim that these are degraded in absence of actin, hence that actin protects them from degradation, they would have to show that the association of MBsomes with lysosomes increases (by IF). They could also show that the decrease of MBsomes in Lantraculin-treated cells is inhibited by lysosomal protease inhibitors and/or bafilomycin.

Fig. 7: The authors conclude that because α V β 3 (and pFAK) is detected in co-localization with MBsomes, this protein signals from the structure. Knock-down experiments, for example, would be required to make a more conclusive statement.

Supp. Fig. 2B: A bright field should be shown to see the cleanness of the preparation. 2C: The Western blot for PDI is of poor quality. Markers for other organelles should also be included. 2D: What tells that the MB in these cells are truly internalized and not simply attached to the surface?

Altogether, the manuscript presents an interesting "story" that is however often poorly supported by the results obtained, which are in several cases lacking statistics (or low levels of significance), or of poor quality (some of the imaging). Furthermore, the study is mostly descriptive. Results presented as showing a functional aspect, such as the involvement of MBsomes in cell proliferation, and MBsome as a signaling organelle that regulates cell proliferation and anchorage-independent growth, are supported by simple experiments providing no mechanistic insights. Most of the study is, in fact, directed towards the characterization of certain aspects of the internalization of MBs (Figs. 4 and 5), providing elements supporting the obvious concept that they are internalized by phagocytosis. Further functional characterization of the molecular mechanisms regulating potential signaling events from the MBsomes (and their direct link to cell proliferation for example) would require more in depth studies.

Reviewer #1:

1. Others have shown that MBs can be internalized (i.e. Crowell et al, 2014, showed loss of MB when entering an acid compartment, and Pohl and Jentsch, 2009, showed colocalization of MB with autophagy components). The data on MB internalization is not particularly convincing. The only data that suggests internalization is the colocalization of MB with CD63, but if MB are in the lysosome how do the authors reconcile that they are still GFP positive since the GFP signal is quickly lost in acidic lysosomes (Fig. 1C, D)??

While others have shown that MBs can be internalized by a daughter cell at the end of telophase, nobody so far has reported that in mammalian unrelated interphase cells can also uptake released MBs, much less that those MBs are retained and can signal. To ensure that MBs are internalized we always stain cells with either phalloidin-Alexa568, CD63 or use cells that express mCherry-CAAX. Then we always take a Z-stack of images to ensure that MBs are inside the cell rather than laying on the plasma membrane surface. To provide an example what we classify as internalized MB we now show an example of single image from the Z-stack as well as 3D renderings from the Z-stack (Supplemental Figure 2D). MDCK and MDA-MB-231 images shown in Supplemental Figure 3F are also just a single image taken from the middle of the Z-stack. Finally, we also replaced images of MBs inside CD63 organelles. New images now clearly show the presence of the “doughnut shape” membranes around the MB (see Figure 1). We also added a new image generated using Nikon SIM super-resolution microscope (see Figure 3D). In this case we “fed” GFP-MBs to the cells expressing mCherry-CAAX. Images clearly show internalized MB that is surrounded by mCherry-CAAX membrane (derived from internalizing cell).

Reviewer is absolutely correct stating that GFP get quickly quenched in acidic environment, potentially leading to an under-estimation of internalized MBs. We tested that by co-staining internalized MBs with anti-GFP and anti-CD63 antibodies (see Supplemental Figure 3A). All internalized MBs that are positive for CD63 and anti-GFP also still have MKLP1-GFP fluorescence. The fact that we still see GFP fluorescence inside CD63 membranes is additional evidence that MBs retain their own membranes, thus preventing acidification of the MB inside and quenching of MKLP1-GFP (for model see Figure 8).

2. Other concern is the preparation of MB. Does it contain exosomes or microvesicles? Other contaminants in these preparations could be responsible for the effect on proliferation. Moreover, the control is “unfed” cells. Is it possible to isolate MB that are not functional (mutant of MKLP1 or others, “empty liposomes?”) to use as controls? Also, in the physiological setting, how many MBs do cells usually take up and what percentage of cells internalize MBs? Is it

possible that exogenously added MBs are not reflecting what is occurring in the natural culture condition?

While we appreciate reviewer concern, we feel that even if MB prep has some micro-vesicle/exosome contamination it is not likely to be the reason for increased proliferation. In all our functional assays we incubated cells with GFP-MBs and then either visually (by microscopy) or flow sorted cells in GFP positive and negative pools. Thus, GFP negative pool (our “unfed” control) was also exposed to everything that may be contaminating MB prep. We re-wrote parts of the manuscript to make that more clear.

As far as we know there are no “not functional” MBs. Empty liposomes also will not really work as a good control since they can easily fuse with plasma membrane while MBs do not. Besides, it would be very hard to visualize them just after the fusion preventing identification of cells that internalized liposomes. Thus, instead in revised manuscript we used Alexa488-tagged E.coli BioParticles (since HeLa do not uptake acrylic beads) as an additional control. Consistent with our model, internalization of these BioParticles had no effect on cell proliferation (Supplemental Figure 2C).

In tissue culture we typically see that about ~15-20% cells uptake MBs and it is reasonably common to see one cell can having 2-3 MBs. To test whether exogenously added MBs are reflecting what is occurring in the natural culture condition is a bit harder. There have been few published studies that shown some tissue culture cells have multiple post-abscission MBs. That is fully consistent with the idea that MB uptake and MBsome formation are important in vivo. Since this manuscript established a concept of MBsome further work will be needed to fully dissect MBsome function in vivo (and we are very interested in pursuing that).

3. Other concern is that authors don't describe what statistical tests are used to analyze the data (and in some figures [Fig. 1D, 2B, 2C, Supl. Fig.1E], there is no statistical analysis).

Where statistical analysis was missing we have repeated experiments and now show statistical analysis for all data in all Figures. As suggested we also added brief description of statistical analysis in Material and Method section. Please note that due to extensive re-organization of the figures, some of the specific data panels may not match specific numbers listed in reviewer comment #3.

Authors show the effect of MB on Hela cells and other cell types on cell proliferation, but it would be useful to also show this in other cell types.

We now also show MBsome-induced proliferation data using MDA-MB-231 cells.

4. Figure 1 (and Suppl. Fig. 1)

Authors do not provide enough convincing data to show that MB are internalized and not just “bound”. Is it possible to label MB with pHRodo or Cypher5E to definitively show that they are internalized and reach the lysosomal compartment?

As we wrote in response to concern #1, in all experiments we always image cells using Z-stack and some sort of plasma membrane marker to ensure that MBs are internalized. Additionally, we wash the cells after feeding and before additional 24-48 hour incubation. Finally, we added new higher quality images of MBs surrounded with CD63(Figure 1C) or mCherry-CAAX images (Figure 3D).

Authors argue that MB evade lysosomal degradation because a fraction of MBs is not associated with CD63. This seems to be an overstatement and needs to be shown convincingly.

In this revised manuscript we now include new MB and CD63 co-staining images. It is now clear that some of post-mitotic MBs are surrounded by CD63 positive membrane (see Figure 1), while others do not. We also followed internalized MBs for up to 72 hours. Thus, some MBs are clearly not rapidly degraded at least for 24-72 hours. We do agree, however, that statement “MBs evade lysosomal degradation” is a bit too strong, thus we re-wrote our conclusions by simply stating that even after 72 hours post-feeding some MBs remains not associated with CD63-positive membranes.

Suppl. Fig. 1B shows different GFP intensity: which fraction was sorted?

We apologize for the confusion. Low GFP fraction was designated as GFP- and high GFP fraction was designated as GFP+. We did not use middle fraction for RNAseq analysis. We have changed the figure legend to make that clear.

Suppl. Fig. 1C shows that –GFP-MB cells still have MB (~ half of +GFP+MB). This is a concern when no (or little) differences are detected in gene expression of stemness markers (or Ki67 in Suppl. Fig. 1E; is this significantly different?)

Since flow cytometry did not allow us complete separation between cells with or without MBs it is possible that the small increase in “stemness” markers could not be detected. Thus, we were careful not to state that MBs are not involved in regulating stemness.

All data shown in Supplemental Figure 1 is significantly different. We also would like to point out that we did RNAseq analysis using three independently generated libraries for each experimental condition. We edited figure legends and Method and Materials Section to make that clear.

5. Figure 2

Authors claim that MB induce cell proliferation and anchor-independent growth. As stated above, the concern here is the MB preparation, and what are the contaminants? Is there a better control than “unfed” cells? Does PS-BD or RGD inhibit the effect on proliferation?

As we described in response to concern#2 in all experiments our “unfed” cells are isolated from the pool of cells that were exposed to the MB prep. These cells simply did not uptake the MB, but were exposed to all possible contaminants. We also would like to point out that our purification conditions should easily separate MBs (that sediment readily due to their 1-3 um size) from any microvesicles or classical exosomes that are much smaller.

Fig. 2A shows that the effect on proliferation (24h) is only observed on cells with >2 MB. This should also be taken into consideration in Fig. 2E (7d) where 2 populations are observed (+GFP+MB group): one with low number/colony and another with high number/colony. Is it possible that the increased proliferation is a product of more cargo, more cellular content, change in cell volume? Will artificial beads stimulate proliferation? Again, a better control is necessary (not just “unfed” cells).

We appreciate reviewers concerns about possible indirect effects of general uptake (rather than MB-specific uptake) on proliferation. As suggested we added two additional controls. First, we tested whether uptake itself (of any large extracellular object) can lead to increased proliferation. As suggested, we tried acrylic beads. It turned out that HeLa cells do not really uptake acrylic beads, even if they are opsinized (despite multiple attempts). Thus, instead we used BioParticles-Alexa488. BioParticle-Alexa488 are fluorescently tagged and heat-killed E.coli (Thermofisher) and HeLa cells readily internalized them (Supplemental Figure 2C). Consistent with our proposed model, internalization of BioParticles (instead of MBs) did not have any effect on cell proliferation (Supplemental Figure 2C).

We also wondered whether naturally occurring sub-population of faster dividing HeLa cells may also be better at internalizing extracellular objects. If that is the case, then faster dividing cells simply internalize more MBs instead of MBs stimulating proliferation. Thus, we added new control to test that (see Figure 2D). In this control we plated cells as single cells and let them grow for 72 hours to form colonies. Cells were then “fed” with purified MBs for 3 hours and then the engulfment ability of cells in large colonies (faster dividing cells) and small colonies (slower dividing cells) were compared (all colonies were on the same coverslips). As now shown in new data (Figure 2D), cells in large colonies do not uptake MBs faster. Actually, they seem to uptake MBs a bit less efficiently. In any case, that is consistent with our model that MB internalization stimulates proliferation, rather than faster dividing cells uptake more MBs.

Why do the authors show different time points for different cells? Are there

differences in the kinetics between cell types – this needs to be explained. Also, MDA-MB-231 cells don't show an actin coat (Suppl. Fig. 2D) as HeLa cells (Fig. 6A). This needs to be reconciled with the effect on proliferation?

Yes, HeLa and MDA-MB-231 cells do show different proliferation kinetics. Additionally, we used two different assays. In one case (for HeLa cells) we plated individual cells at very low density on glass bottom dishes and followed colony formation from a single cell with or without the MB. In other case (for MDA-MB-231 cells) we flow sorted cells into GFP-positive and GFP-negative pools and plated all of them on plastic dish. The total number were then counted. These cells did not have to grow from single cell into colonies, thus they tended to grow much faster.

We also added new data showing actin coats in MDA-MB-231 cells. We also quantified the percentage of HeLa cells that actually have actin coats and added additional HeLa MBsomes with actin coats images.

Fig. 2D: Are these the FACS sorted cells used in experiments in Fig. 2F? Do the cells conserve the MB after 14 days of culture?

Yes, we used the same FACS sorting for both experiments. I think it is very unlikely that our original GFP-tagged MB survived after 14 days. It is much more likely that the MB signaled at the very beginning stimulating anchorage-free survival and initial proliferation of embedded individual cells. We managed to follow uptaken MBs for as long as 3-4 days. While after that MBs are likely degraded, they have already affected survival and initial proliferation of cells. We edited our text to make that more clear.

6. Figure 3

The authors test whether the MB are “internalized” via fusion or engulfment and show that MB are surrounded by membrane protrusions. These data doesn't show either fusion or engulfment nor internalization. Also, there are membrane protrusions where no MB are found. What is the percentage of MB associated to protrusions? Again, if most MB are associated with the lysosomal marker CD63, why GFP is not quenched by the acid pH? If MB are targeted to the lysosomes, Cypher labeling might help to show this.

We agree with the reviewer that protrusions are very hard to visualize using light microscopy. To better visualize these protrusions we used high-resolution 3D tomography. All new data is shown in Figure 3 and Supplemental Movies 1 and 2. The tomography analysis clearly shows the presence of protrusions, thus fully consistent with our initial model. Significantly, tomography shows that at the point of contact between protrusions and MBs one can observe a coat-like electron dense layer. The presence of this layer further supports the involvement of MB-specific recognition machinery rather than simply non-specific charge-dependent sticking to plasma membrane.

The fact that GFP is not quenched by acidification in CD63-containing organelles (see new data in Supplemental Figure 3A) is another indication that MBs are surrounded and retain their own membrane, thus protecting GFP (tagged to internal MB protein MKLP1) from quenching. To further analyze that we stained internalized MBs with anti-GFP antibody found that GFP and anti-GFP colocalize almost in 100% of the observed cases.

7. Figure 4

Authors show colocalization of actin protrusions and MB. What is the effect of an actin inhibitor (such as EHT1864) or Cytochalasin D in the binding and "internalization" of MB? (LatA is used in Fig. 6 after internalization to evaluate how MB evade degradation).

As suggested we tested Cytochalasin D as well as Rac1 inhibitors. Both of them decreased MB internalization (see new data in Figure 4D).

8. Figure 5

Authors show nicely that both isolated MB and "endogenous" MB (Fig. 5G) externalize PtdSer using PS-BD. Does the control GST show any non-specific binding? The staining with the control construct is not shown and the GST alone seems to have a mild effect on MB binding (Fig. 5B). It would be useful to add another staining for PtdSer, such as Annexin V.

As suggested we added GST-only staining as a control (see Figure 5F). We have attempted to use anti-annexin V antibody to stain MBsomes but were unable to find one that works for immunofluorescence imaging.

Authors study the effect of blocking PtdSer (PS-BD) and integrins (RGD) on the binding (3h) and internalization (24) of MB. Again, authors do not provide convincing data to show that the MB are really internalized (the different time points is not enough).

As we described in answers to previous concerns/suggestions, we always ensure that MBs are internalized by doing Z-stack imaging and co-staining with plasma membrane markers?

Fig. 5E: Why RGE and GST controls are not shown? Do they have any effect? Have authors use antibodies anti- $\alpha V\beta 3$ as an alternative block?

As suggested we show GST and RGE controls (see Figure 5B and D). We have considered using anti- $\alpha V\beta 3$ to block internalization. However, the way the experiment is set up, that you use a lot of antibodies and would be prohibitively expensive.

9. Figure 6

Here, authors evaluate a possible role of the actin "coats" to evade MB degradation.

What is the fraction of MB with actin coats?

As suggested we counted the number of internalized MBs (after 24 hours) that has clearly detectable actin coats, which turned out to be 48%. The new data is now stated in the text of the manuscripts.

LatA “reduces” the number of MBsomes. Are more MB associated with CD63 after LatA treatment? A reduced number of MBsomes is not conclusive.

As suggested we also analyzed whether LatA treatment increases the number of MBs associated with CD63. The new data is shown in Figure 6E (left bar graph) and does show a statistically significant increase in MBs associated with late endosomes/lysosomes.

LatA doesn't have an effect on proliferation on either –GFP-MB or +GFP+MB cells. The 3rd and 4th bars are not different.

As expected LatA did not really have any effect on basic proliferation (-GFP-MB bars). LatA did eliminate GFP-MB induced increase in proliferation since +GFP-MB cells briefly treated with LatA are not statistically different from –GFP-MB cells anymore. We edited figure and text to make that more clear.

Effect is analyzed after 36h without LatA? Is LatA reversible?

Yes, LatA is fully reversible. Cells regain their actin cytoskeleton within 1 hour of the wash-out of the drug.

Is this cell type specific? MDA-MB-231 cells do not show actin coats (Suppl. Fig. 2D). Please explain.

No, actin coats are not cell-specific. Just like in HeLa cells, in MDA-MB-231 cells one can observe actin coats in sub-population of MBs (presumably the ones that are not being degraded). We added images of actin coats in MDA-MB-231 cells (see Supplemental Figures 2E and 3B).

Figure 7

May be I'm not interpreting the photos correctly, but why are the integrins only detected on the MB and not on whole Hela cells? These MBs were derived from Hela cells? The co-staining with pFAK and pEGFR are correlative, they do not necessarily show a requirement for gene expression and/or MB internalization. Authors will need to use mutants or inhibitors to show that these pathway is involved.

The images shown are the optical lane at the middle of the cells (taken from Z-stack). As the consequence one does not see integrins that are located at the bottom of the cell.

We do agree with the reviewer that this study does not fully “nail down” the signaling of EGFR and integrins from the MB. Unfortunately, these are very tricky experiments to do. Using inhibitors, knock-downs or mutants of EGFR and integrins has a very dramatic effect on cell survival and proliferation even in cells that were not “fed” MBs. As a consequence, it is impossible to interpret these assays. The only way would be to inhibit EGFR or integrins specifically at the MB, but we could not figure a way to actually do that. Besides, we feel that specific signaling pathways are not really a focus of this paper. We are very interested in better defining and analyzing MB-signaling pathways and will be focusing on that in our future studies. The main goal of this study is to define post-abscission MB uptake and identify MBsomes as novel signaling organelle that affects cell proliferation.

Do PS-BD or RGD block pFAK and pEGFR localization with MB?

Yes, but indirectly. Since PS-BD and RGD act early in the pathway, they actually block MB internalization. As the result we cannot test whether pFAK and pEGFR still localize to MBsomes.

Minor points

Figure 5D should say binding not uptake

Corrected.

Typos:

“my microscopy” line 618, page 14

“liposomes” line 640, page 15

Corrected.

Reviewer #2:

1. Many crucial experiments (e.g. Figure 2A, 2B, 2D, 2E, 2F, 2G) are based on the comparison between cells incubated with or without purified MBs. Unfortunately, there is no real characterization of the degree of purity of the added MBs, which are isolated from the extracellular medium of cultured cells. MB proteomic analysis (Suppl. Table 2) actually demonstrates that the preparation contains many contaminants while many key MB proteins are lacking (Cep55, PRC1, KIF20A...). How pure is this MB prep? Is it contaminated with cells (as suggested by the proteomic analysis), exosomes, intracellular/plasma membranes, RNAs etc? As a better negative control, the authors should treat the cells with MB-negative purifications from cells that do not produce MBs, which would include all the potential contaminants.

We cannot really generate MB-negative purifications since all mammalian cells generate MBs during cell division. Any mammalian cell line that we and others have tested (HeLa, MDA-MB-231, MCF10A, MDCK, MCF7 just to name the few) always release at least some of the MBs in the media. We have added additional control, such as effect of internalizing BioParticles (since HeLa do not uptake beads), although this experiment controls for possible effect on proliferation by internalizing large external objects.

While we appreciate reviewer concern about possible effect of contaminations, we feel that even if MB prep has some micro-vesicle/exosome contamination it is not likely to be the reason for increased proliferation. In all our functional assays we incubated cells with GFP-MBs and then either visually (by microscopy) or flow sorted cells in GFP-MB positive and negative pools. Thus, GFP-MB negative pool (our “unfed” control) was also exposed to everything that may be contaminating MB prep. So both pools were exposed to the same possible contaminants, but -GFP-MB cells did not internalize MBs while +GFP-MB cells did. We re-wrote parts of the manuscript to make that more clear.

2. Once the authors resolve the contamination issue, the physiological relevance of the experiments should be addressed. Indeed, it seems that HeLa cells exceptionally contain more than 2 MBs per cell. One issue of the manuscript is that there is no indication of the number of MBs added per cell in order to observe a biological effect. What is the mean number (and number distribution) of exogenous MBs in Figure 2A, 2B, 2D, 2E, 2F, 2G? How does it compare to the distribution of MB numbers per cell in an untreated population?

In “fed” cells we observe HeLa cells having 2 internalized MBs quite often (about 25% of MB-containing interphase cells). Cells with 3-4 MBs can also be observed but less often, only in about ~5% of cells (see example of cell with 3 MBs in Supplemental Figure 1C). In contrast, while we occasionally see one post-abscission MB in untreated cell, it is very uncommon to see 2 and almost never more than 2. In part that is likely due to that fact that typically HeLa cells are grown in large volume of media, thus diluting released MBs. For “feeding” experiments we typically add 90,000 MBs per 200,000 cells in 0.5 mls of media (thus ratio of MBs/cell is 0.45). We observed all reported effects under these conditions. As suggested by a reviewer we now added this information to Materials and Methods section. Please note, that published reports from several laboratories have shown that in tissue or tumor section it is pretty common to see some cells containing more than 3 MBs, although it was never understood how these post-abscission MBs are accumulated and what functional consequences of this MB accumulation may be. We believe that this study helps to answer that question.

It is also very likely that the number of added GFP-positive MBs, based on GFP-positive signals, is underestimated by the authors. Since GFP is quenched in acidic compartments, antibodies against GFP should be systematically used to

address the localization and number of added MBs per cell (e.g. Figure 1D, 3D). How many MBs detected with anti-GFP are found after 3h, 24h and 48h?

As suggested we have used anti-GFP antibodies to stain cells that internalized GFP-MBs (24 hours post-feeding). We found that anti-GFP and GFP overlapped hundred percent (see new data in Supplemental Figure 3A). Thus, we do not underestimate number of internalized GFP-MBs. We also think that the lack of GFP quenching in CD63-positive organelles is another indicator that MBs retain their own membrane even after being internalized. This MB-associated membrane (remnant from dividing cell) is clearly visible in our new tomography analysis of MB uptake (see figure 3). This MB membrane would also protect GFP-MKLP1 signal from acidification-dependent quenching since MKLP1 is internal MB protein.

3. Related to this question, how solid is the evidence that MBs are internalized and not merely at the cell surface after 24h of incubation? The authors must demonstrate that the washing step indeed removes non-internalized MBs. Z-stacks should also be systematically provided when relevant (Figure 3, Figure 6, Figure 7). For instance, the MBs in Figure 6A, 6C, 7A seem to be located at the cell surface (with an actin phagocytic cup described in *C. elegans* and in cultured cells), which is not consistent with MBs in MBsomes.

Reviewer is absolutely correct that the use of Z-stack imaging is a key to determine whether MBs are on the surface or inside the cell. To ensure that MBs are internalized we always stain cells with either phalloidin-Alexa568 or use cells that express mCherry-CAAX. Then we always take a Z-stack of images to ensure that MBs are inside the cell rather than laying on the plasma membrane surface. All images shown are single images taken from the Z-stack at the MB level. Due to space limitations we cannot show all images from the Z-stack for every experiment. However, to provide an example what we classify as internalized MB we now show an example of single image from the Z-stack as well as 3D renderings from the Z-stack (Supplemental Figure 2D). MDCK and MDA-MB-231 images shown in Supplemental Figure 3F are also just a single image taken from the middle of the Z-stack. We also replaced images of MBs inside CD63 organelles. Now images now clearly show the presence of the “doughnut shape” membranes around the MB (see Figure 1 and Supplemental Figure 3A). Finally, we add new better quality and resolution image (taken using SIM) of cells expressing mCherry-CAAX that internalized GFP-MB (Figure 3D). Image clearly shows inside the cell and surrounded by the membrane derived from internalizing cell.

4. The notion of signaling by the MBsome (Figure 7) is interesting but it is not sufficiently supported by the data. First, the authors use anti- β 3 integrin and anti-EGFR antibodies to suggest that MBsomes are signaling vacuoles. However, EGFR was already found in the proteome of intact MBs (suppl. Table2). Therefore, are

we looking at EGFR present at the plasma membrane of MBs or at the surface of the MBsome? The authors should demonstrate that EGFR/pEGFR staining (and potentially $\beta 3$ integrin staining) is not detected on intact MBs or on MBs at the cell surface.

As suggested we stained purified post-abscission MBs with anti-pFAK, anti-pEGFR and anti- $\alpha V\beta 3$ antibodies. The new data is now presented in Supplemental Figure 3B and clearly shows that there is no signal of pFAK and pEGFR in purified intact MBs, thus consistent with our model that MB internalization and MBsome formation is needed for MB-associated pEGFR and pFAK signal. We did see very weak $\alpha V\beta 3$ signal. However, it is clearly much weaker than the one observed in MBsomes (see Figure 7).

The authors suggest that CD63- MBs escaped degradation and function as MBsomes that could thus signal. If 87% of MBs are in $\alpha V\beta 3$ -positive MBsomes (Figure 7A), most MBs should be in CD63-negative compartments, which apparently contradicts the results presented in Figure 1D (only 30% of the MBs are CD63-). Why is there such a difference? The same holds true for the EGFR.

It is difficult to compare directly percentages of MBsomes that are CD63 positive and percentage of MBsomes that contain integrins and EGFR. Typically, early endosomes and phagosomes slowly mature to become late endosomes and eventually fuse with lysosomes. During this maturation cells deliver CD63 and other late endosome proteins, thus slowly acidifying organelle until it is ready to fuse with lysosome. This process is now well described for signaling endosomes containing activated EGFR. It is well established that while signaling endosomes slowly mature to become late endosome, EGFR still signals while present in signaling endosomes already containing CD63. We think MBsomes work in very similar fashion and that integrin and EGFR continues to signal (at least for a while) in MBsomes already containing CD63. Furthermore, actin coats are likely slowing down this maturation process rather than completely inhibiting it. We edited text to make that more clear.

Again, these results might be more consistent with the idea that many MBs are not internalized even after 24h and that we are looking at the EGFR/ $\beta 3$ -integrin of the MB plasma membrane. Ideally, correlative EM should be provided (showing that the EGFR or integrins are activated at the surface of MBsomes).

It certainly possible (actually quite likely) that EGFR and Integrins may already start to signal when MB engages cellular plasma membrane to be internalized. That actually would be very consistent with our general model. We are certain that after 24 hr vast majority of EGFR and Integrins containing MBs are already internalized (based on Z-stacking). We also agree that CLEM followed by immune-EM would be great. However, these are very difficult experiments that really can only be done in a very few labs in the world. Our University has no technical capability of that. We have added tomography

images of MB in the process of being bound and internalized using facility in UC Boulder. However, this facility cannot do CLEM. The tomography images shown are the result of heroic efforts by my student Eric Peterman (first author on manuscript) to scan through a lot of cells to find this one example.

Second, it is crucial to demonstrate that the effect of MBs on cell proliferation is exerted via signaling from $\beta 3$ integrin or EGFR. To this aim, the RNAseq experiment requested in the previous point should be compared with $\beta 3$ integrin-depleted or EGFR-inhibited cells.

These are hard experiments to interpret. Integrins and EGFR has direct effect on cell proliferation even without internalized MBs. Inhibiting or depleting EGFR and/or integrin would have wide-ranging effects on function and the whole transcriptome that will be impossible to correlate to MBsome signaling. Additionally, "nailing-down" exact signaling pathways is not really a focus of this manuscript. In this study we are identifying MBsomes as a novel signaling organelle that regulate proliferation as well as characterizing the basic machinery mediating MB uptake. We do at the end propose possible pathways of MBsome signaling, but it will take an additional substantial effort (which we are planning to do) to clearly define signaling pathways, especially since there are several other RTKs that are known to cluster with integrins (such as $TGF\beta$ receptor) and could also mediate MBsome signaling.

Specific issues

1. Figure 3D: the absolute number of MBs (see also Major concerns #2) should be provided, not the % Total. How many exogenous MBs per cell are CD63+/CD63- after 3h, 24h and 48h? How many are eventually degraded? How efficient is MB internalization?

As suggested we now provide absolute MB numbers in addition to %. We also have shown that about 25% of MB-containing cells has more than one MB. Finally, we also calculated the efficiency of MB internalization. In "feeding" assays we usually add ~0.4 MB per cell and get about ~15-20% of cells to uptake MBs. We added all these new numbers to the text of the manuscript.

2. Figure 4 is incremental. It is already known in the literature that MBs are internalized by a phagocytic-like mechanism relying on F-actin in mammalian cells and in *C. elegans* (LC3-dependent, Rac1-dependent phagocytosis). The abstract should be revised accordingly. This Figure would be more informative if the authors could show that this phagocytic event is also Rac1-dependent in mammalian cells. The localization of CCN1, EDIL3 and/or MFG-E8 around MBs should also be provided.

Reviewer is absolutely correct that that MB internalization dependency on actin was already shown in HeLa and C.elegans cells (and we cite all these studies). However, we

felt that it is important to test that in this particular scenario, since in all published studies MB internalization was done by one of the post-mitotic daughter cells followed by immediate degradation. In our study MB is uptaken by unrelated cell and is kept for 1-2 days during which time it continued signaling. As suggested we also add new data showing that MB uptake is dependent on Rac1 (see figure 4D). This does correspond nicely to previous C. elegans data that showed the Rac1 ortholog was required for MB engulfment.

As suggested we have tried to show that CCN1, EDIL3 and MFG-E8 are present around MBs. Unfortunately, there are no good antibodies that works for IF. Besides, determining which one (are perhaps all of them) of these proteins are required for MB uptake is not really then main interest of this study. What we wanted to show is that PS is required for this uptake. Which exact PS-receptor (likely multiple receptors) mediates this internalization will probably be also dependent on the cell type, thus is not really a key point in this manuscript.

3. Figure 6: the notion of the actin “coat” is not well established. The only clear coat is seen in Figure 6A, but I believe that this is rather a phagocytic event. What is the percentage of MBs displaying this actin staining? In Figure 6C, there is no visible coat. Why would it not be present here like in Figure 6A?

The image shown in Figure 6A is very unlikely to be a phagocytic event since MB is already present in the middle of cell (based on Z-stacking). As we mentioned in the manuscript, actin coats appear to be very dynamic, constantly polymerizing/depolymerizing around the MBsome. Indeed, similar highly actin coats have been observed on other endosomes and phagosomes where they also play a role in determining endocytic sorting and endosomal fate. These are also typically very dynamic and hard to visualize/detect on all endosomes. To ensure that actin coats are not just a property of HeLa cells we added images of MBsome actin coats in MDA-MB-231 cells (see Supplemental Figure 2E and Supplemental Figure 3B).

As suggested we also determined the percentage of internalized MBs (after 24 hours) that has clearly visible actin coats, which turned out ~48% (58 randomly picked internalized MBs analyzed). The new data is now stated in the text of the manuscript.

4. Figure 5: The GST alone and RGE alone controls should be included. In addition, the number of internalized MBs detected with anti-GFP would be very informative.

We do now show GST alone and RGE controls in Figure 5. As suggested we also stained all internalized MBs with anti-GFP antibody and show that co-localization between anti-GFP and GFP is essentially 100%.

5. Figure 6D: why is this an “MBsome”? Is it CD63-?

In this particular case we did not stain for CD63 since these are time-lapse images of cells expressing mCherry-LifeAct and fed with purified GFP-MBs. We call it MBsome because it is still present after 24 hours of feeding and is internalized (based on Z-stacking). The main purpose of this image was to show that MBsomes are quite motile inside the cells, presumably due to differential and dynamic actin polymerization on different sides of the MBsome, thus pushing it around the cell.

Figure 6F: are the conditions +GFP with DMSO vs. LatA statistically different?
LatA treatment is known to inhibit MB phagocytosis. Could it be that the observed effect is the consequence of decreased MB internalization.

+GFP/DMSO and +GFP/LatA are not statistically different. LatA treatment diminished (although did not completely blocked it) the effect of MBsomes on proliferation, thus making it not statistically different from both -GFP/DMSO and +GFP/DMSO). We also do not think that this is effect on MB uptake, since short LatA treatment was administered 24 hours after "feeding". By then the vast majority of MBsomes are already internalized (based on Z-stacking).

In addition, does the addition of LatA promote the recruitment of CD63/Lamp1 around MBs and their subsequent degradation?

As suggested we analyzed the effect of LatA on recruitment of CD63 around MBs. New data is shown in Figure 6E (right bar graph) and does demonstrate that actin depolymerization enhances lysosomal MB degradation.

6- Figure 2A: pictures should be provided. Does the Ki67 staining scale with the number of exogenous MBs per cell? What is the average number of "2+ GFP-MBs"?

As suggested we added Ki67 images to Figure 2A and are now also including (in the text) the percentage of cell with 2 or more MBs (about 25%).

7- Figure 2B: FACS analysis of the cell cycle (including M phase) should be provided. Are the - vs. + GFP-MBs statistically significant? Are the - vs. + GFP-MBs statistically significant?

As suggested we added statistical analysis and yes, -GFP-MB and +GFP-MB are significantly different (see figure 2B). We cannot do the FACS analysis on cell cycle since during any feeding experiment only portion of cells internalized MBs and it is very tricky (due to low GFP signal) to gate the analysis based on presence of MBs. We have attempted this but could never get a very reliable gating to be included in the manuscript. So, instead we did microscopy analysis by counting what fraction of cells are

in M phase (based on staining with anti-acetylated tubulin and DAPI staining) in +GFP-MB and -GFP-MB populations. Please note that all cells were exposed to MB prep (and any possible contaminants) and then washed and incubated for another 24 hours. Cells were then categorized (from the same coverslip) whether they are -GFP-MB or +GFP-MB based on presence of internalized MBs.

8. Figure 2D: Error bars and statistics should be provided.

All data that did not had statistical analysis in previous version of the manuscript has been repeated and error or standard deviation bars have been added.

9. Lines 69-70 page 3: cite the relevant literature in *C. elegans* and mammalian cells.
Lines 182-184 page 5: this has already been addressed in *C. elegans* and mammalian cells.

Lines 206-208 page 6: this has already been addressed in *C. elegans* and mammalian cells. Confirmation is not a problem but previous literature should be cited.

As suggested we now include all these references.

10. Figure 6D: the correlation is not obvious. Please present single channels and arrows in the snapshot.

Due to space limitation in figure, especially to accommodate all the new data that we are adding to revised manuscript, we cannot add single channels of all time-lapse stills. To illustrate what MBs some actin coat looks like we did add single channel mCherry-LifeAct inset for that last time-point. Arrow in the inset points to the black space where the MB is located. One can clearly see an actin coat around it. Box marks the part of the image that was used for the inset.

11. In the title, “post-abscission midbody” or “post-cytokinetic midbody” would be more appropriate.

Changed as suggested to “post-abscission midbody” in the title and in the text.

Reviewer #3:

1. Fig. 1C: The labeling for CD63 in the lower panel cannot be presented as representing a membrane labeling around a MB. In such cases, one should see a narrow rim around MB. What is shown here looks more like blobs. To address this important point in a more convincing way, the authors should express a fluorescently-tagged version of CD63 and follow phagocytosis in these cells.

Our apologies for the poor quality image. It was replaced by the better image that clearly shows a CD63 ring around the MB. Another image of CD63 “ring” is also not shown in Supplemental Figure 3A.

Fig. 1E does not present statistics. Is that because the measurement was performed only once? If so, it should be repeated and statistics presented.

As suggested experiment was repeated a few more times and bars now show means and standard deviations. Please note that 1E is now 1D.

2. Fig. 2A: Images of the expressing cells should be shown. 2B, C and D: Why is there no error bars and/or statistics?

As suggested we added Ki67 images to Figure 2A. Where it was needed we also repeated experiments and performed statistical analysis. Thus, all quantitative data shown in the manuscript now has statistical analysis.

3. Fig. 3A and B: The legend indicates an incubation of 3h. What is the “1h” marking on the image refers to?

It is 1 hour post-feeding to allow cells to start internalizing the MBs. Sorry for the confusion. We edited figure legend to make that more clear.

Fig. 3C: I am not convinced by the image stating that a MB is surrounded by a membrane (arrow). Such claim would be more substantiated by EM analyses.

Unfortunately, it is very hard to do EM analysis of internalized MBs using conventional thin-section EM or tomography since it is difficult to find the internalized MBs by randomly sectioning fed cells. The only way to do it is to use CLEM. But even that is very tricky and can really only be done by a very few labs and our University is not equipped to do CLEM. In this revised manuscript we do add tomography analysis of the MB in the process of being internalized due to heroic effort by graduate student (Eric Peterman, first author on the manuscript). We also added a new and higher resolution (using Nikon SIM microscope) of mCherry-CAAX expressing cells that contain internalized GFP-MB (Figure 3D). Image clearly shows that MB is inside the cell and is surrounded by the membrane coming from internalizing cell.

Fig. 3E: The images are of very poor quality and the CD63 labeling is not convincing. Idem for Fig. 3F (super pixilated). Furthermore, in Fig. 3F, I find the claim that MBsomes have 2 membranes inconclusive. Again, EM analyses would potentially solve this issue.

Sorry for poor quality images. They have been removed. CD63 and MB co-staining is now shown in Figure 1 and Supplemental Figure 3A using much better quality images. To show double membrane images using EM analysis is virtually impossible due to issues of

finding the internalized MB in 3D cell, unless University has CLEM capabilities (which we do not, see answer above). The idea that MBsomes can have two membranes can be partially resolved by our newly added tomography of the MB in process of being uptaken. One can clearly see that MB is surrounded by its own membrane (remnant from division) and that cell's plasma membrane is making contacts with MB membrane during MB internalization. This would be consistent with MB uptake by phagocytosis-like mechanism (which would create two membranes) rather than simple fusion of MB membrane and cell's plasma membrane.

4. Fig. 4: This figure is more like a parenthesis in the study to show that the phagocytosis of MBs involves actin. The requirement of actin in all form of phagocytosis is well known and documented.

Reviewer is absolutely correct that that MB internalization dependency on actin was already shown in HeLa and C. elegans cells (and we cite all these studies). However, we felt that is important to test that in this particular scenario, since in all published studies MB internalization was done by one of the post-mitotic daughter cells followed by immediate degradation. In our study MB is uptaken by unrelated interphase cell and is kept for 1-2 days during which time it continued signaling. In this revised manuscript we also add new data showing that MB uptake is also dependent on Rac1 (see figure 4D).

5. Fig. 6A-D: This part of the manuscript describes the association of MBsomes to actin structures (i.e. coats and tails). Fig. 6E: The authors observe a decrease in the number of MBsomes upon treatment with Lantraculin (an actin depolymerizer), concluding that the decrease is due to degradation. Although this could be the case, what they simply observe is a decrease in MBsome number. To claim that these are degraded in absence of actin, hence that actin protects them from degradation, they would have to show that the association of MBsomes with lysosomes increases (by IF). They could also show that the decrease of MBsome in Lantraculin-treated cells is inhibited by lysosomal protease inhibitors and/or bafilomycin.

As suggested we now include new data (see Figure 6E, right bar graph) that shows and increase in CD63 associated MBs after LatA treatment.

6. Fig. 7: The authors conclude that because integrins (and pFAK) is detected in co-localization with MBsomes, this protein signals from the structure. Knock-down experiments, for example, would be required to make a more conclusive statement.

The integrin knock-down studies would be very difficult to interpret. Upon adhesion to the substrate integrins also signal from focal adhesion sites. Furthermore, this focal adhesion signaling is very important for cell survival and proliferation. Cytokinesis itself is also affected by integrins. Any integrin knock-down would lead to multiple cell-wide

effects that will be virtually impossible to relate to MBsomes.

7. Supp. Fig. 2B: A bright field should be shown to see the cleanness of the preparation. 2C: The Western blot for PDI is of poor quality.

As suggested, we have taken bright field image and nothing can be seen on it, since even MBs are too small to visualize by bright-field. Consequently, due to limited space we decided not include them.

8. Supp. 2C: Markers for other organelles should also be included.

We and others previously published that MBs contain numerous post-Golgi organelles, including lysosomes, recycling and early endosomes. Consistently with previously published MB proteome (as well as MB proteome in this manuscript) MBs contain numerous endocytic markers, such Rab8, Rab11, Rab35. Actually, we and others previously published that the presence of Rab11 and Rab35 at the midbody are required for successful completion of cytokinesis. For these reasons we believe that western blotting MB fraction with other post-Golgi markers would ne be very informative.

9. Supp. 2D: What tells that the MB in these cells are truly internalized and not simply attached to the surface?

To ensure that MBs are internalized we stain cells with either phalloiding-Alexa568 or use cells that express mCherry-CAAX. Then we always take a Z-stack of images to ensure that MBs are inside the cell rather than laying on the plasma membrane surface. To provide an example what we classify as internalized MB we now show an example of single image from the Z-stack as well as 3D renderings from the Z-stack (Supplemental Figure 2D). MDCK and MDA-MB-231 images shown in Supplemental Figure 3F are also just a single image taken from the middle of the Z-stack. Similarly, all images shown in manuscript figures are single optical plane images taken from Z-stack.

Reviewers' comments:

Reviewer #2 (Remarks to the Author):

In this revised manuscript, the authors clarified a number of points. However there are still important issues that should be addressed and that are listed below. Whether the intracellular compartments that contain the internalized midbodies (MBsomes) are really where proliferation signals come from, as well as the nature of these signals, remains unclear.

Rebuttal Major issue #1:

The new experiment with BioParticles is puzzling (Supp. Fig. 2): in this set up, the cells do not proliferate at all (less than 1 division in 3 days for HeLa cancer cells!). The same is true for Fig. 2F. Why is this?

The BioParticle experiment does not address a fundamental question in this study: could it be that cells that better internalize midbodies are precisely cells that naturally divide faster (not the other way around, as presented in the revised manuscript)? The following experiment would directly answer this question. First, treat cells with GFP-MB and sort the GFP+ vs. GFP- populations. Then, isolate clones after 2-3 weeks and test 1) whether there is a difference of GFP-MB internalization in the clones arising from the 2 initial populations (i.e. do cells retain their phagocytic properties over time?) and 2) whether the cells derived from each population display differences in their proliferation rate (in the absence of added midbodies). If the latter were true, then it invalidates the idea that the added MBs directly control cell proliferation (since it is unlikely that the initial boost provided by the MBs added 3 weeks before would last so long).

Rebuttal Major issue #2:

Please provide the distribution of cells with 0-1-2-3 etc. midbodies in cells treated with GFP-MBs vs untreated cells in Suppl. data.

"Please note, that published reports from several laboratories have shown that in tissue or tumor section it is pretty common to see some cells containing more than 3 MBs".

Please provide references. How common is it really? Where has this been quantified?

It is very surprising that 100% of MBs detected with anti-GFP are green (indicating that GFP is not quenched by acidic pH). Given the model proposed in Figure 8, a fraction of MBs are not in MBsomes and should be detected by anti-GFP but quenched. In addition, this does not fit with published data reporting that endogenous MBs are engulfed, quenched and degraded, even when they are generated by non-sister cells. The authors should investigate whether the endogenous GFP-MBs in the cell line that they used for purifying MBs are quenched after engulfment using video microscopy (in live cells to avoid potential artifacts of pH/fixation), as reported by others. If they were, it would mean that purified, exogenous MBs do not behave as endogenous MBs, which would be problematic. In addition, what is the percentage of exogenous GFP-MB that are positive for anti-GFP and Lamp2? Is it also 100%?

Rebuttal Major issue #4:

"It certainly possible (actually quite likely) that EGFR and Integrins may already start to signal when MB engages cellular plasma membrane to be internalized. That actually would be very consistent with our general model."

In the manuscript, the authors claim that signaling occurs from internal MBsomes ("a novel signalling organelle") but now agree that signaling could arise from the cell surface. Where does

signaling come from? This is a fundamental point that should be addressed. Is proliferation also increased if MBs are put in contact with cells but washed out before internalization? This is important for clarifying the whole concept of a signaling MBsome.

From the initial Review: "Second, it is crucial to demonstrate that the effect of MBs on cell proliferation is exerted via signaling from β 3 integrin or EGFR. To this aim, the RNAseq experiment requested in the previous point should be compared with β 3 integrin-depleted or EGFR-inhibited cells."

Although the Reviewer appreciates that it is a difficult question, the notion that pEGFR is detected around the MBs is the essence of the notion of a MBsome proposed in this manuscript. Without this experimental evidence, it is still unclear whether it is really a signaling organelle as proposed (see also above).

Major Issue #2 of the initial Review:

"Once the contamination issue is resolved, a RNAseq experiment should be carried out on cells treated with control vs. MB+ purifications. This would provide a global characterization of the variation in gene expression upon MB treatment and would be much more informative than the experiment presented in Suppl. Table 1."

This has not been addressed nor commented on in the Rebuttal letter. This is an important experiment that will demonstrate whether MBs indeed induce the global transcriptional changes suggested in Suppl. Table 1.

Rebuttal Specific issue #2: without requested staining or tagged proteins, Fig. 4E remains speculative.

Rebuttal Specific issue #3:

The notion of "actin coats" is unclear. This should be better defined. New Suppl. Fig. 2E show tails or comets but not a surrounding coat, as depicted in the new Figure 8.

Rebuttal Specific issue #4:

The GST and RGE controls alone are still missing in Fig. 5E (the review specifically referred to Fig.5E).

Rebuttal Specific issue #5:

It is important to give a definition for MBsomes.

Rebuttal Specific issue #7:

"We cannot do the FACS analysis on cell cycle since during any feeding experiment only portion of cells internalized MBs and it is very tricky (due to low GFP signal) to gate the analysis based on presence of MBs."

I understood that all functional experiments (e.g. rate of proliferation of clones) were based on FACS sorting of MB-positive cells. Please, explain why cell cycle analysis is not possible.

FACS analysis of the cell cycle would solve the very puzzling results presented in Fig.2B (30% of cells treated with MBs are in mitosis). How is this possible? Either cells proliferate 3 times faster

(which is obviously unlikely), or cells spend 3 fold more time in M-phase, or a combination of these two possibilities, or cells are not in M-phase. Somehow, this has to be experimentally addressed.

In Suppl. Fig. 4A, please display the same number of cells in left and right panels for proper comparison.

Additional comments:

Figure 1B: the model presented in the right panel should be removed, since the literature in *C. elegans*, *D. melanogaster* and mammalian cultured cells argues against this view proposed many years ago. Figure 1B should instead describe what is already known for endogenous MBs and present the two reported possibilities: release into the medium or engulfment.

"Importantly, pre-incubation of cells with Rac1 inhibitor or cytochalasin D blocked MB internalization (Figure 4D)," (p.6) is an overstatement.

Reviewer #3 (Remarks to the Author):

This is a much better version of the manuscript than the original version submitted. The authors have improved the quality of the figures and added the needed experiments to make statistical claims. However, although interesting, the study in its present form is mostly descriptive, as the demonstration that signalling effectively occurs from MBsomes is still missing. The most important aspect that is still missing from this study is convincing data showing that signalling effectively occurs from the MBsomes. This was also asked by the first reviewer.

This lowers my overall enthusiasm for the manuscript.

We would like to thank reviewers for constructive comments (especially reviewer #2). In this revised manuscript we addressed **all** concerns and completed **all** proposed additional experiments. It substantially improved the manuscript. Additionally, we also included new electron microscopy data (Fig. 3D) that was not requested by either of the reviewer. However, since in first revision reviewers wanted more evidence that MBs are internalized and are containing within membrane-bound organelle, we did CLEM analysis to further prove that. This new EM data nicely complements already existing fluorescence microscopy analysis. The point-by-point responses to proposed experiments/concerns are listed below.

Reviewer #2

In this revised manuscript, the authors clarified a number of points. However there are still important issues that should be addressed and that are listed below. Whether the intracellular compartments that contain the internalized midbodies (MBsomes) are really where proliferation signals come from, as well as the nature of these signals, remains unclear.

We have added all requested experiments. Hopefully, they will now address all of the concerns, including the nature of the signaling from the MBsomes. We appreciate insightful reviewers comments and strongly believe that addressing them substantially improved the manuscript. Detailed list of added experiments and edits are listed below.

Rebuttal Major issue #1:

The new experiment with BioParticles is puzzling (Supp. Fig. 2): in this set up, the cells do not proliferate at all (less than 1 division in 3 days for HeLa cancer cells!). The same is true for Fig. 2F. Why is this?

In both of these experiments cells were seeded at very low density (less than 5% confluency). In that case we can count cells as they form colonies. We do that to ensure that we can easily identify, follow and count cells originating from a individual flow-sorted cells that contain GFP-tagged MBs. It is quite common that cells plated at very low densities initially divide much slower. Thus, we always see less robust proliferation during first 24 hours after plating. Importantly, MB-containing cells (but not Bioparticle containing cells) do not have this lag in proliferation (see Figure 2F), suggesting that MBs may enhance cell clonogenic properties (for both HeLa and MDA-MB-231 cells). That would be consistent with suggestions in previous publications that MBs can induce “stemness”. However, while this is very interesting observation, we felt that we do not have enough data to support claim that MBsomes stimulate “stemness”, thus we did not add discussion about this possibility to the manuscript. We can certainly add it if reviewer feels that it would enhance the study.

The BioParticle experiment does not address a fundamental question in this

study: could it be that cells that better internalize midbodies are precisely cells that naturally divide faster (not the other way around, as presented in the revised manuscript)? The following experiment would directly answer this question. First, treat cells with GFP-MB and sort the GFP+ vs. GFP- populations. Then, isolate clones after 2-3 weeks and test 1) whether there is a difference of GFP-MB internalization in the clones arising from the 2 initial populations (i.e. do cells retain their phagocytic properties over time?) and 2) whether the cells derived from each population display differences in their proliferation rate (in the absence of added midbodies). If the latter were true, then it invalidates the idea that the added MBs directly control cell proliferation (since it is unlikely that the initial boost provided by the MBs added 3 weeks before would last so long).

We completed requested experiment (see new Supplemental Figure 3). Fully consistent with our proposed model, cells derived from both populations (1) did not display differences in MB internalization and (2) did not exhibit different proliferation rates once MBs were degraded.

Rebuttal Major issue #2:

Please provide the distribution of cells with 0-1-2-3 etc. midbodies in cells treated with GFP-MBs vs untreated cells in Suppl. data.

“Please note, that published reports from several laboratories have shown that in tissue or tumor sections it is common to see some cells containing more than 3 MBs”.

Please provide references. How common is it really? Where has this been quantified?

As requested we added data showing distribution of cells with MBs. To that end we counted cells incubated with purified GFP-MBs and analyzed distribution of cells with MBs. Total, ~26% cells internalized MBs. From all GF-MB containing cells 55.8% had 1 MB, 22.1% had 2 MBs, 12.6% had 3 MBs and finally 9.6% had 4 MBs (n=95). We added these data to the text of the manuscript.

We do agree with the reviewer that we are only beginning to understand what role MB accumulation play in vivo. However, there are some publications that does demonstrate MB accumulation (in cells other then HeLa) and are cited in this manuscript. One of them comes from Steve Doxsey's lab (NCB, 2011, PMID:21909099). In this paper authors show that multiple MB (per cell) accumulation can be observed in stem cells of mouse seminiferous tubes, neuron progenitor cells in mouse brain and mice hair follicle stem cells. We also do see MB accumulation in LGR-positive cells in mice intestinal crypts (unpublished data). We also previously have shown (PMID:29296475) that multiple MBs accumulate in spheroids grown from squamous cell carcinomas (SCCs) and that accumulation of these post-mitotic MBs increase SCC invasive properties. We edited the text of the manuscript to make that more clear.

It is very surprising that 100% of MBs detected with anti-GFP are green (indicating that GFP is not quenched by acidic pH). Given the model proposed in Figure 8, a fraction of MBs are not in MBsomes and should be detected by anti-GFP but quenched. In addition, this does not fit with published data reporting that endogenous MBs are engulfed, quenched and degraded, even when they are generated by non-sister cells. The authors should investigate whether the endogenous GFP-MBs in the cell line that they used for purifying MBs are quenched after engulfment using video microscopy (in live cells to avoid potential artifacts of pH/fixation), as reported by others. If they were, it would mean that purified, exogenous MBs do not behave as endogenous MBs, which would be problematic. In addition, what is the percentage of exogenous GFP-MB that are positive for anti-GFP and Lamp2? Is it also 100%?

As suggested we performed time-lapse microscope on engulfed GFP-MBs. The data is now shown in Supplemental Figure 2G. We think the cause of confusion was that all data so far was done in fixed cells. As new time-lapse images show, once cell decide to degrade MBs, the GFP disappears (GFP quenched and MB degraded) very quickly. Thus, at steady state in fixed cells that vast majority of detectable MBs are the ones that are not being degraded (thus not acidified). Indeed, CD63 marker will also label late endosomes that are not acidified yet. Thanks to great suggestion by reviewer, the time-lapse movies now clearly show that MBsomes are eventually acidified and degraded. That certainly is consistent with previous work as well as our proposed model. We have analyzed 14 internalized MBs by time-lapse and have shown that in 16 hours post-internalization 43% of MBs are quenched/degraded and 57% are retained. It is very likely that eventually all MBsomes will get acidified and all MBs will get degraded, thus terminating MBsome signaling.

Rebuttal Major issue #4:

“It certainly possible (actually quite likely) that EGFR and Integrins may already start to signal when MB engages cellular plasma membrane to be internalized. That actually would be very consistent with our general model.”

In the manuscript, the authors claim that signaling occurs from internal MBsomes (“a novel signalling organelle”) but now agree that signaling could arise from the cell surface. Where does signaling come from? This is a fundamental point that should be addressed. Is proliferation also increased if MBs are put in contact with cells but washed out before internalization? This is important for clarifying the whole concept of a signaling MBsome.

The possibility that EGFR signal at the plasma membrane (upon MB binding) as well as from the MBsome are not really mutually exclusive. It is now becoming clear that tyrosine kinase as well as integrin signaling often required both, plasma membrane and intracellular organelle signaling component. Indeed, it is now well-established that to get full EGFR signaling itinerary EGF receptors need to signal at plasma membrane (usually short term) as well as after internalization and targeting to the signaling endosomes (long term). Typically, this EGFR

signaling is terminated after 3-6 hours due to eventual targeting to lysosome. Recently, similar behavior was also reported for integrin receptors by Johanna Ivaska's laboratory. She has shown that integrin signaling from plasma membrane as well as from specialized endosomes (termed endoadhesome) are both required for full scale integrin signaling. We hypothesize that MBsomes may function in similar manner. Thus, MBs may initiate signaling as early as during initial binding to integrins at the plasma membrane. However, we do know that internalization and signaling from MBsome is also crucial since many of signaling effects (such as increase in proliferation) can be observed as late as 24-48 hours after internalization. Furthermore, this MBsome induced proliferation can be blocked if MBs are degraded (after treatment with latrunculin A, see Fig. 6E-F) or after addition of EGFR inhibitors (newly added data). We hypothesize that EGFR clustering with integrins as well as extended protection of EGFR from degradation is what mediates MBsome signaling. It is worth noting that new studies from several labs now demonstrates that integrin clustering with TK receptors is also needed for full complement of signaling. MBsomes may also serve the role in allowing this clustering. We have edited the manuscript to make these points more clear.

From the initial Review: "Second, it is crucial to demonstrate that the effect of MBs on cell proliferation is exerted via signaling from β 3 integrin or EGFR. To this aim, the RNAseq experiment requested in the previous point should be compared with β 3 integrin-depleted or EGFR-inhibited cells."

Although the Reviewer appreciates that it is a difficult question, the notion that pEGFR is detected around the MBs is the essence of the notion of a MBsome proposed in this manuscript. Without this experimental evidence, it is still unclear whether it is really a signaling organelle as proposed (see also above).

As suggested in the revised manuscript we test directly whether EGFR signaling is required for MB-induced increase in proliferation (see Figure 7G). To that end, we incubated HeLa cells with GFP-MBs and then flow sorted them into GFP-MB positive and negative populations. Equal numbers from both populations were then plated and incubated for 48 hours with or without EGFR inhibitor. We picked 48 hours since that is when most of MBsome effect seem to be (based on our other proliferation data). Cells were then washed to remove EGFR inhibitor and grown for another 48 hours followed by cell counting. As is shown in new data, the presence of MBsomes again stimulated HeLa proliferation. However, the MBsome-induced increase in final cell number was completely blocked by the presence of EGFR inhibitors (Fig. 7G). That is fully consistent with the proposed hypothesis that EGFR signals from MBsomes to increase proliferation.

Major Issue #2 of the initial Review:

"Once the contamination issue is resolved, a RNAseq experiment should be carried out on cells treated with control vs. MB+ purifications. This would provide a global characterization of the variation in gene expression upon MB treatment and would be much more informative than the experiment presented in Suppl.

Table 1.”

This has not been addressed nor commented on in the Rebuttal letter. This is an important experiment that will demonstrate whether MBs indeed induce the global transcriptional changes suggested in Suppl. Table 1.

As suggested we completed a second RNAseq analysis. In this case, we incubated HeLa cells with purified GFP-MBs. 24 hours later, cells were then flow-sorted in GFP-MB positive and negative populations. Please note, that even if purified GFP-MBs have some contaminants, both cell populations would have been exposed to it. The cells were then processed for RNAseq analysis. The new RNAseq data is now shown in Table 4 and Table 5. Consistent with our hypothesis, the presence of internalized GFP-MBs again led to an increase in subset of mRNAs that are involved in cell proliferation. Importantly, many of the mRNAs identified in our first RNAseq, were again increased in our newly completed transcriptome analysis. Note that new RNAseq analysis also showed increase in Cyclin B and Cyclin D (see Supplemental Table 5), both genes that are known to be upregulated at transcriptional level in response to EGFR-induced proliferation. This is more evidence that MBsomes may function, at least in part, via EGFR.

Rebuttal Specific issue #2: without requested staining or tagged proteins, Fig. 4E remains speculative.

As requested we now add western blots (Supplemental Figure 2B) as well as images (Supplemental Figure 3E) to show that MFG-E8 is present in purified MBs (presumably binding to outer MB membrane). Interestingly, in images one can see that MFG-E8 is enriched in what appears to be cell plasma membrane and MB contact sites. That is fully consistent with the hypothesis that cells forms protrusions (see tomography in Figure 3B) and binds MBs by recognizing PS/MFG-E8 on MB membrane.

Rebuttal Specific issue #3:

The notion of “actin coats” is unclear. This should be better defined. New Suppl. Fig. 2E show tails or comets but not a surrounding coat, as depicted in the new Figure 8.

Our sincere apologies for the confusion. We never meant to imply that MBsomes are always surrounded by continuous actin coat. In fact, as reviewer point out, in most cases MBsome-associated actin is very dynamic, forming patches and tails. These types of actin patches and tails were also implicated in “protecting” phagosomes from fusion with lysosomes (PMID:19638408), although molecular machinery governing this process remains to be fully defined. Our data demonstrate that actin likely plays similar role on MBsomes. Thus, actin “coat” is probably not the best term. We replaced it with term “actin patches”.

Rebuttal Specific issue #4:

The GST and RGE controls alone are still missing in Fig. 5E (the review specifically referred to Fig.5E).

As requested we added controls. See new Figure 5E bar graph. Consistently with out hypothesis, GST alone or RGE had no effect on MB internalization.

Rebuttal Specific issue #5:

It is important to give a definition for MBsomes.

Since internalized MBs remain separated from cytosol by membrane (actually two) and behaves/signals in a manner similar to signaling endosomes we called them MB-associated signaling endosomes or MBsomes. We edited text to make that more clear.

Rebuttal Specific issue #7:

“We cannot do the FACS analysis on cell cycle since during any feeding experiment only portion of cells internalized MBs and it is very tricky (due to low GFP signal) to gate the analysis based on presence of MBs.”

I understood that all functional experiments (e.g. rate of proliferation of clones) were based on FACS sorting of MB-positive cells. Please, explain why cell cycle analysis is not possible.

FACS analysis of the cell cycle would solve the very puzzling results presented in Fig.2B (30% of cells treated with MBs are in mitosis). How is this possible? Either cells proliferate 3 times faster (which is obviously unlikely), or cells spend 3 fold more time in M-phase, or a combination of these two possibilities, or cells are not in M-phase. Somehow, this has to be experimentally addressed.

In Figure 2B we used anti-acetylated tubulin as a marker to identify the cells that did not complete abscission. Thus, “cells in mitosis” will include the cells that are still connected with intracellular bridge. These cells already flattened out and can sometimes be counted as “interphase” unless you stain cells for midbody marker. Since final abscission does take 1-2 hours (in HeLa cells), it is not that uncommon to see 10-30% of cells that still not underwent abscission. However, as suggested, we did complete cell cycle analysis and now show data (that is fully consistent with our model) in Supplemental Figure 3G, that also includes raw flow data traces. The analysis was repeated 3 times and shown data (in bar graph) are the means and standard deviations.

In Suppl. Fig. 4A, please display the same number of cells in left and right panels for proper comparison.

To calibrate gating we used fewer cells than for actual experiment, thus we cannot display data differently. Please note that in Supplemental Figure 4B we show a data that further validates our gating and sorting efficiency.

Additional comments:

Figure 1B: the model presented in the right panel should be removed, since the literature in *C. elegans*, *D. melanogaster* and mammalian cultured cells argues against this view proposed many years ago. Figure 1B should instead describe what is already known for endogenous MBs and present the two reported possibilities: release into the medium or engulfment.

We do agree with the reviewer that multiple evidence seem to point out that in C. elegans and HeLa (although for HeLa it is less clear) MBs appear to be cut on both sides and released into the media or internalized by one of the daughter cells. Thus, we do also think that MB release rather than “inheritance” (by only cutting on one side) is probably most likely abscission outcome. That actually fits very well with our MBsome model. However, we do not think that we can completely rule out that in some cases and in some cell lines MBs can be inherited. We actually do occasionally see that even in HeLa cells. Thus, trying to be completely non-bias we feel that we need to present both mechanisms at the beginning of the manuscript. We did, however, added to this new revised manuscript version a couple sentences that states that majority of recent data seem to support double-cut and release model (see Introduction).

“Importantly, pre-incubation of cells with Rac1 inhibitor or cytochalasin D blocked MB internalization (Figure 4D),” (p.6) is an overstatement.

Thank you for pointing it out. We do agree that term “blocked” is a bit to strong. In new manuscript it was replaced with term “decreased” that represents data better.

Reviewer #3

This is a much better version of the manuscript than the original version submitted. The authors have improved the quality of the figures and added the needed experiments to make statistical claims. However, although interesting, the study in its present form is mostly descriptive, as the demonstration that signalling effectively occurs from MBsomes is still missing. The most important aspect that is still missing from this study is convincing data showing that signalling effectively occurs from the MBsomes. This was also asked by the first reviewer. This lowers my overall enthusiasm for the manuscript.

In response to second set of comments from reviewer #2 we have added numerous new experiments, including the ones testing the hypothesis that MBsomes signal to regulate cell proliferation. We strongly believe that these new experiments makes a much stronger case that MBsomes are actually novel signaling organelles. The new manuscript now have following evidence for that:

(1) we show that EGF-EGFR complex is present in MBsomes for at least 24 hours after MB internalization; (2) we show that MBsomes colocalize with anti-phospho-EGFR staining, indicating that EGFR is activated; (3) we demonstrate that EGFR inhibitors block MB-induced proliferation; (4) we show that $\alpha V\beta 3$ integrin complex is present in MBsomes; (5) we also demonstrate that phospho-FAK staining colocalizes with MBsomes indicating that integrin complex is activated; (6) our data demonstrates that induction of MB degradation blocks increase in cell proliferation and (7) finally we show that MB internalization increases anchorage-independent growth and proliferation in at least two different cell types. We strongly feel that all these evidence makes a strong case that MBsomes can, in fact, signal. We also hope that with all these data as well as new functional assays reviewer #3 does not think that manuscript "is mostly descriptive" any longer.

We also wanted to point out that the concept of MB uptake and signaling as double-membrane organelle has never been proposed before. Thus, the main focus of this manuscript is to identify MBsome as a novel signaling organelle and to define the machinery of MB uptake and MBsome formation, rather than to identify the specific signaling pathways. Obviously, description of these signaling pathways will be a very interesting next step. However, much more work will be needed to fully categorize the itinerary that mediate MBsome signaling. Once this manuscript is published (hopefully in Nature Communications) it will lay a foundation for us and others to do just that.

Reviewers' comments:

Reviewer #1 (Remarks to the Author):

The authors have adequately addressed the various concerns of the reviewers. While there are always larger experiments to pursue, this work is sufficiently novel that I support the publication of this work.

Reviewer #4 (Remarks to the Author):

The authors thoroughly addressed the comments of reviewer #2. Nevertheless, some questions remain, and some of the experiments bring up new questions that counteract the current interpretations.

1) Major issue of signaling directionality: The authors now nicely demonstrate that the EGFR and integrin signaling components are present around the internalized midbody. This can be interpreted as enhancement of proliferation signaling originating from the internalized post-abscission midbody. Another publication (Lujan et al. *Commun. Integr. Biol.* 2017) also demonstrated signaling coming from isolated midbodies. Although these MBs were isolated in a different procedure by cell lysis of MDCK cells and the signaling was in the polarity context, this finding would support the possibility of signaling coming from the midbody. On the other hand, the authors also nicely characterized the nature/structure of the midbody-containing endosomes, revealing the presence of a membrane originating from the abscission and one from the uptake. They demonstrate that the signaling components originate from the "outer membrane", i.e. the one from the uptake, not from the post-abscission midbody. Furthermore, they characterized the actin patches surrounding the internalized midbodies, and mention that 48% of internalized midbodies are surrounded by these patches. In their rebuttal they write that "These types of actin patches and tails were also implicated in "protecting" phagosomes from fusion with lysosomes (PMID:19638408), although molecular machinery governing this process remains to be fully defined." Thus, the actin coating happens in only half of the cases of uptake. Could this be by chance? How does this compare to other phagosomes (data from the literature?) and midbodies from other cell types? These would be better controls than dead bioparticles.

Therefore, while possible it is unclear if in this case the midbody itself is the origin of the signaling, or if the uptake leads, possibly by chance, to a coating that induces the signaling by increasing the internalization of the signaling components of proliferative HeLa cancer cells. Therefore, naming the midbody-containing endosomes as signaling organelles is still an over-interpretation. The authors need to take all possibilities of interpretation into account and write a more balanced discussion.

2) The authors show uptake of HeLa-derived midbodies in two more cell types, but do not show that their conclusions on the function of the internalized midbodies are as broad as they claim. They only use HeLa cells for that, and only HeLa-derived midbodies. In their Peterman and Prekeris 2016 publication they wrote: "it is possible that midbodies derived from different cells can contain different proteins and lipids" and "it is likely that MBs isolated from different cell types may have distinct composition and properties". Thus, in order to claim that midbody-containing endosomes are in general signaling organelles is again an over-interpretation, as this could be HeLa-specific.

3) I think that a deeper, less selective discussion on previous studies would be required. In Ettinger et al., a paper that the authors cite, it was shown by live imaging that "HeLa cells almost always (>90% of the cases) showed midbody-retention". And "Midbodies retained by HeLa cells that persisted for longer than one cell cycle were not released during the next cell cycle but rather contributed to the presence of multiple midbodies retained by these cells, as observed previously by other investigators". This should be mentioned and put into context with the current study, particularly because HeLa cells are used here. Also, Crowell et al. reveal that midbodies are taken

up not only by daughter cells, as the authors mention here, but also distant cells through an actin-dependent phagocytosis-like uptake mechanism, similar to the one described here. Crowell et al also used HeLa cells, and they also describe the appearance of an actin ring around the midbody. This has been understated in the current manuscript.

The authors mention that they "performed an analysis of the functional consequences of retaining post-abscission MBs" (lines 441, 442). Given what I wrote in point 1, this is not true as the internalized midbody-containing endosomes have the membrane coat of the host cell, and the differences should be discussed, especially in the background of the literature. If the authors think that a retained midbody has the same properties (for example membrane structure) as an internalized one, then they need to show that.

Nevertheless, this is a very interesting manuscript. I particularly appreciate the phosphatidylserine-switch and "midbody maturation" revealed in this study. This makes it more likely that this is an active uptake similar to described ones, not random phagocytosis, and supports a role of the internalized midbody rather than a "phagocytic" enrichment of proliferation signaling components by chance as outlined above.

Reviewer #4:

The authors *thoroughly addressed* the comments of reviewer #2. Nevertheless, some questions remain, and some of the experiments bring up new questions that counteract the current interpretations.

We are happy with the conclusions reviewer #4 regarding addressing reviewer #2 comments (in previous submission). As stated, the reviewer has some additional new comments/suggestions. While these suggestions were not listed in reviewer #2 comments, we have incorporated all of them in this newly revised manuscript. All changes in the manuscript are marked in yellow. For point-by-point changes see below.

1) Major issue of signaling directionality: The authors now nicely demonstrate that the EGFR and integrin signaling components are present around the internalized midbody. This can be interpreted as enhancement of proliferation signaling originating from the internalized post-abscission midbody. Another publication (Lujan et al. Commun. Integr. Biol. 2017) also demonstrated signaling coming from isolated midbodies. Although these MBs were isolated in a different procedure by cell lysis of MDCK cells and the signaling was in the polarity context, this finding would support the possibility of signaling coming from the midbody.

We are delighted to see that the reviewer was impressed with our data regarding EGFR and integrin signaling from the midbody. We also agree that recent paper by Lujan and colleagues is supportive of our idea that post-mitotic MBs can actually signal (in their case in the context of regulating polarity). Thus, we added citation and mention of this paper to our manuscript (Discussion section; page 11). This paper also demonstrates that MB-dependent signaling is not Hela specific since all the studies were done using MDCK cells.

On the other hand, the authors also nicely characterized the nature/structure of the midbody-containing endosomes, revealing the presence of a membrane originating from the abscission and one from the uptake. They demonstrate that the signaling components originate from the “outer membrane”, i.e. the one from the uptake, not from the post-abscission midbody.

Again, we are delighted that reviewer is happy with our studies regarding nature of MB signaling.

Furthermore, they characterized the actin patches surrounding the internalized midbodies, and mention that 48% of internalized midbodies are surrounded by these patches. In their rebuttal they write that “These types of actin patches and tails were also implicated in “protecting” phagosomes from fusion with lysosomes (PMID:19638408), although molecular machinery governing this process remains to be fully defined.” Thus, the actin coating happens in only half of the cases of uptake. Could this be by chance? How does this compare to other phagosomes (data from the literature?) and midbodies from other cell types? These would be better controls than dead bioparticles. Therefore, while possible it is unclear if in this case the midbody itself is the origin of the signaling, or if the uptake leads, possibly by chance, to a coating that induces the signaling by increasing the internalization of the signaling components of proliferative HeLa cancer cells. Therefore, naming the midbody-containing endosomes as signaling organelles is still an over-interpretation. The authors need to take all possibilities of interpretation into account and write a more balanced discussion.

As suggested by reviewer, to demonstrate that MB-associated actin patches are also present in other cell types (not just HeLa) we have now added new data (see supplemental figure 4B) showing actin patches in MDA-MB-231 and 293T cells (using mCherry-MBs purified from 293T cells). Thus, we now show that MB internalization and actin-patch formation can be observed in at least three different cell types.

We also now have a short discussion regarding the previous observation (with references) of actin-association with phagocytic structures containing either bacteria or internalized beads (see page 9).

As reviewer pointed out only about half of internalized MBs are associated with actin patches. It is not that surprising, since (according to our data) some of the internalized MBs do get degraded and presumably these actin-free MBs are targeted to lysosomal pathways. We also wanted to point out that it is unlikely that 50% association with actin is “by chance”, especially

since short-term treatment of internalized MBs with actin-depolymerizing drugs stimulates MBsome their degradation.

Finally, we also do not think that internalization event itself stimulates signaling, since uptake of beads or bioparticles does not lead to the phenotype observed after MB internalization (see our data). This strongly suggests that MB internalization, rather than induction of generic phagocytosis, is what leads to signal activation. As suggested by the reviewer we expanded discussion to take all interpretations into account (see Discussion section).

2) The authors show uptake of HeLa-derived midbodies in two more cell types, but do not show that their conclusions on the function of the internalized midbodies are as broad as they claim. They only use HeLa cells for that, and only HeLa-derived midbodies. In their Peterman and Prekeris 2016 publication they wrote: “it is possible that midbodies derived from different cells can contain different proteins and lipids” and “it is likely that MBs isolated from different cell types may have distinct composition and properties”. Thus, in order to claim that midbody-containing endosomes are in general signaling organelles is again an over-interpretation, as this could be HeLa-specific.

In this manuscript we show that internalized MBs stimulate proliferation and anchorage-independent growth using two different cell lines, namely HeLa and MDA-MB-231. Thus, it is clearly not a HeLa cell specific phenomenon. This idea is also supported by recent publication demonstrating that post-mitotic MBs also regulate MDCK cell polarity. All these data are clearly consistent with the idea that MBsomes may be present and function in different cell types. However, we do agree that we did not show that MBsomes are general and widely-used signaling structure in all cells. Thus, as suggested by reviewer, we eliminated sentences stating that MBsomes are general signaling organelle (see Discussion section, page 11 and last sentence in Abstract).

3) I think that a deeper, less selective discussion on previous studies would be required. In Ettinger et al., a paper that the authors cite, it was shown by live imaging that “HeLa cells almost always (>90% of the cases) showed midbody-retention”. And “Midbodies retained by HeLa cells that persisted

for longer than one cell cycle were not released during the next cell cycle but rather contributed to the presence of multiple midbodies retained by these cells, as observed previously by other investigators". This should be mentioned and put into context with the current study, particularly because HeLa cells are used here. Also, Crowell et al. reveal that midbodies are taken up not only by daughter cells, as the authors mention here, but also distant cells through an actin-dependent phagocytosis-like uptake mechanism, similar to the one described here. Crowell et al also used HeLa cells, and they also describe the appearance of an actin ring around the midbody. This has been understated in the current manuscript.

We completely agree with the reviewer that MBs appear to be accumulated via two distinct pathways, one by inheritance and one by internalization. This is the reason why we included a MB accumulation schematic in Figure 1B. As suggested in current reviewer comments, we further expanded the discussion on this issue to ensure that we discuss MB accumulation in "less selective" fashion (see Introduction section, page 2). We also included mention that Crowell et al study has observed internalization of release MBs by distant cells (see Introduction section, page 2).

The authors mention that they "performed an analysis of the functional consequences of retaining post-abscission MBs" (lines 441, 442). Given what I wrote in point 1, this is not true as the internalized midbody-containing endosomes have the membrane coat of the host cell, and the differences should be discussed, especially in the background of the literature. If the authors think that a retained midbody has the same properties (for example membrane structure) as an internalized one, then they need to show that.

We fully agree with the reviewer that inherited and internalized MBs likely lead to different functional consequences. As reviewer pointed out, that is likely due to the presence of membranous envelope in internalized MBs. We actually have a discussion on this topic in the Discussion section (page 11, paragraph 4). To make that more clear, we expanded this discussion (page 11, paragraph 4) as well as restated the sentence quoted by the reviewer. Now it says "performed an analysis of the functional consequences of internalized post-abscission MBs".

Nevertheless, this is a very interesting manuscript. I particularly appreciate the phosphatidylserine-switch and “midbody maturation” revealed in this study. This makes it more likely that this is an active uptake similar to described ones, not random phagocytosis, and supports a role of the internalized midbody rather than a "phagocytic" enrichment of proliferation signaling components by chance as outlined above.

Once again, we are happy with the overall conclusions by the reviewer that this is a very interesting manuscript and hope that incorporating all suggested text changes as well as one more experiment (using MBs purified from 293T cells) will now make this manuscript suitable for publication in “Nature Communications”.

REVIEWERS' COMMENTS:

Reviewer #4 (Remarks to the Author):

The authors have addressed my comments thoroughly. It is a very interesting and exciting work, and now discussed in a more balanced way.